# Synaptic basis of a sub-second representation of time in a neural circuit model

A. Barri [1] ✉, M. T. Wiechert[1], M. Jazayeri [2,3] & D. A. DiGregorio [1] ✉

Temporal sequences of neural activity are essential for driving well-timed behaviors, but the underlying cellular and circuit mechanisms remain elusive. We leveraged the well-defined architecture of the cerebellum, a brain region known to support temporally precise actions, to explore theoretically whether the experimentally observed diversity of short-term synaptic plasticity (STP) at the input layer could generate neural dynamics sufficient for sub-second temporal learning. A cerebellar circuit model equipped with dynamic synapses produced a diverse set of transient granule cell firing patterns that provided a temporal basis set for learning precisely timed pauses in Purkinje cell activity during simulated delay eyelid conditioning and Bayesian interval estimation. The learning performance across time intervals was influenced by the temporal bandwidth of the temporal basis, which was determined by the input layer synaptic properties. The ubiquity of STP throughout the brain positions it as a general, tunable cellular mechanism for sculpting neural dynamics and fine-tuning behavior.

The neuronal representation of time on the sub-second timescale is a fundamental requisite for the perception of time-varying sensory stimuli, generation of complex motor plans, and cognitive anticipation of action[1–4]. But how neural circuits acquire specific temporal contingencies to drive precisely timed behaviors remains elusive. A progressive increase in firing rate ("ramping") towards a threshold can represent different elapsed times by altering the slope of the ramping behavior. Elapsed time can also be encoded by a population of neurons that fire in a particular sequence ("time cells")[5–8]. Sequential synaptic connections between neurons (synfire chains) can explain the neural sequences representing bird song[9] and contribute to time delays necessary to cancel self-generated sensory stimuli in the electrosensory lobe of mormyrid fish[10]. Temporal dynamics of neural population activity can also be reproduced by training recurrent neural network models[11–13]. Nevertheless, the search for a candidate mechanism for generating a temporal reference (biological timer) for neural dynamics is an ongoing challenge.

Short-term synaptic plasticity (STP) is the rapid change in synaptic strength occurring over tens of milliseconds to seconds that is thought to transform presynaptic activity into distinct postsynaptic spike patterns[14]. Depression and facilitation of synaptic strength can act as low-and high-pass filters, respectively[15], and synaptic depression can mediate gain modulation[16,17]. Network models of neocortical connectivity exhibit improved temporal pattern discrimination when augmented with STP[18]. Within recurrent neural networks, the long timescales of cortical synaptic facilitation provide the substrate for working memory[19]. Finally, low-gain recurrent network models that include STP also show enriched neural dynamics and generate neural representations of time[20]. However, experimental evidence of STP-dependent circuit computations is rare and is associated mainly with sensory adaptation[21].

The cerebellar cortex is a prototypical microcircuit known to be important for generating temporally precise motor[22] and cognitive behaviors[23–26] on the sub-second timescale. It receives mossy fibers (MFs) from various sensory, motor and cortical areas. MFs are thought to convey contextual information and converge onto granule cells (GCs), the most numerous neuron in the brain. The excitatory GCs project onto the inhibitory molecular layer interneurons and Purkinje cells (PCs). PCs, being the sole output neurons of the cerebellar cortex, inhibit neurons in the deep cerebellar nuclei. According to the Marr-

[1]Institut Pasteur, Université Paris Cité, Synapse and Circuit Dynamics Laboratory, CNRS UMR 3571 Paris, France. [2]McGovern Institute for Brain Research, Massachusetts Institute of Technology, Cambridge, MA, USA. [3]Department of Brain and Cognitive Sciences, Massachusetts Institute of Technology, Cambridge, MA, USA. ✉e-mail: alessandrobarri@gmail.com; david.digregorio@pasteur.fr

Albus-Ito model of cerebellar cortical circuit computations, precisely timed Purkinje cell activity can be learned by adjusting the synaptic weights formed by GCs with differing activity patterns[27,28]. This largely feed-forward circuitry has been proposed to learn the temporal contingencies required for prediction from neural sequences across the population of GCs within the input layer[29]. The synapses between MFs and GCs are highly variable in their synaptic strength and STP time course[30]. Therefore, we hypothesized that STP of MF-GC synapses could be used as internal timers for a population clock within the cerebellar cortex to generate neural dynamics necessary for temporal learning.

To elaborate this hypothesis, we modeled the cerebellar cortex as a rate-based two-layer perceptron network that includes realistic MF-GC connectivity and STP dynamics. The model reproduces learned PC activity associated with a well-known temporal learning task: delay eyelid conditioning[31]. The timescales of STP determined the temporal characteristics of the GC population activity, which defined the temporal window of PC temporal learning. The width of PC pauses scaled proportionally with the learned time intervals, similar to experimentally observed scalar variability of the eyelid conditioning behavior[32]. Additionally, we found that STP-driven GC activity was well suited to implement a Bayesian estimator of time intervals[33]. We propose that within neural circuits, dynamic synapses serve as tunable clocks that determine the bandwidth of neural circuit dynamics and enable learning temporally precise behaviors.

## Results

### Cerebellar cortex model with STP

The cerebellar cortex can be modeled as a two-layer perceptron that performs pattern separation of static inputs[27,28,34,35]. Cerebellar models of temporal processing are generally supplemented with additional mechanisms that generate temporally varying activity patterns in the GC layer[10,29,36,37]. To test whether heterogeneous MF-GC STP is sufficient to support temporal learning, we implemented STP of the MF-GC synapse in a simplified cerebellar cortex model, hereafter referred to as $CCM_{STP}$. This model deliberately omits all other potential sources of temporal dynamics. In particular, in most of the simulations presented here, we did not include recurrent connectivity (Fig. 1b). STP was simulated using a parallel vesicle pool model of the MF-GC synapse, similar to ref. 38. It comprises two readily releasable and depletable vesicle pools, synaptic facilitation, and postsynaptic desensitization. To reproduce the observed functional synaptic diversity, we set vesicle fusion probabilities ($p_v$), synaptic pool sizes ($N$), and synaptic facilitation to match the relative strengths, paired-pulse ratios, and transient behaviors across five different types of synapses that were previously characterized[30] (Fig 1a$_2$–a$_6$). Importantly, the longest timescale in $CCM_{STP}$ is associated with a 2 s vesicle refilling time constant of the slow vesicle pool ($\tau_{ref} = 2s$, Fig. 1a$_1$). To capture depression over long timescales[38,39], we introduced a phenomenological parameter ($p_{ref} = 0.6$) that effectively mimics a simplified form of activity-dependent recovery from depression (see Methods).

The $CCM_{STP}$ consisted of firing rate units representing MFs, GCs, a single PC, and a single molecular layer interneuron, i.e., each neuron's activity was represented by a single continuous value corresponding to an instantaneous firing rate. Each GC received 4 MF synapses, randomly selected from the different synapse types according to their experimentally characterized frequency of occurrence[30]. Importantly, we associated different synapse types with different MF firing rates (Fig. 1b, left panels, see Methods). High $p_v$ MF inputs were paired with high average firing rates (primary sensory groups 1, 2) and low $p_v$ synapses with MF inputs with comparatively low average firing rates (secondary/processed sensory groups 3, 4, 5), according to experimental observations[40,41]. We will reconsider this relationship below.

To examine $CCM_{STP}$ network dynamics, input MF activity patterns were sampled every second from respective firing rate distributions shown in Fig. 1b. Each change in MF patterns evoked transient changes in MF-GC synaptic weights, which in turn generated transient GC firing rate

responses that decayed at different rates to a steady-state (Fig. 1c). Similar to experimentally recorded PC responses to sensory stimuli in vivo[42], switches between different MF activity patterns also generated heterogeneous transient changes in the PC firing rate, whose directions and magnitudes were controlled by the ratio of the average excitatory to inhibitory weight (Fig. 1c, bottom). In contrast, when MF-GC STP was removed, the transient GC and PC responses disappeared (Fig. 1d). The amplitude of the firing rate transients increased as the difference from one MF pattern to the next increased, similar to previous theoretical work[16]. Sequential delivery of uncorrelated MF firing patterns in $CCM_{STP}$ (Fig. 1e) generated GC and PC transients with broadly distributed amplitudes (Fig. 1f1,2), which were progressively reduced as the relative change in MF rate decreased (Fig. 1g). Thus, dynamic MF-GC synapses allow both GCs and PCs to represent the relative changes in sensory stimuli.

### Simulating PC pauses during eyelid conditioning

We next explored whether MF-GC STP diversity permits learning of precisely timed PC pauses associated with delay eyelid conditioning, a prototypical example of a cerebellar cortex-dependent learning. In this task, animals learn to use a conditioned stimulus (CS) to precisely time eyelid closure in anticipation of an aversive unconditioned stimulus (US). This eyeblink is driven by a preceding decrease in PC firing rates[31,43] (Fig. 2a). Since the CS is typically constant until the time of the US and a precisely timed eyelid response can be learned even if the CS is replaced by direct and constant MF stimulation[44,45], we modeled CS delivery in the $CCM_{STP}$ by an instantaneous switch to a novel MF input pattern that persists over the duration of the CS (Fig. 2a). Most GC activity transients exhibited a characteristic rapid increase or decrease in firing rate, followed by an exponential-like decay in firing rate (Fig. 2c). In contrast to other models of eyelid conditioning[29], the activity of most GCs in the $CCM_{STP}$ peaked only once, occurring shortly (<50ms) after the CS onset (Fig. 2c). However, the distribution of GC firing rate decay times across the population was highly variable with a fraction of GCs showing decay times to 10% of the transient peak as long as 700 ms (Fig. 2c, d).

To test whether the GC population dynamics could act as a basis set for learning the precisely timed PC firing rate pauses known to drive the eyelid response, we subjected the GC-PC synaptic weights to a gradient descent-based supervised learning rule[46]. The rule's target signal consisted of a square pulse (zero firing rate at a specific time bin) at the designated time of the PC firing rate pause (Fig. 2e, dotted line). In the course of learning, there was a progressive acquisition of a pause in the PC firing rate (Fig. 2e). However, without MF-GC STP, the PC pause did not develop (Fig. 2e, pink). We tested learning of different delay intervals ranging from 25 ms to 700 ms and found that PC pauses could be generated for all delays. The PC pause amplitude and temporal precision (time and width) decreased with increasing CS-US delays (Fig. 2f), reminiscent of the shape of PC simple-spike pauses recorded during eyelid conditioning[31].

Why might the learned PC pause amplitude and temporal precision be reduced for longer CS-US delays? The parameters associated with the learning algorithm (e.g. the number of iterations) are identical for each CS-US delay. The state of the GC population activity, in contrast, changes throughout the CS. Once all GC activity dynamics reach steady-state, temporal discrimination by PCs is diminished, and interval learning becomes impossible. In other words, for temporal learning to be effective, changes in GC firing rates must be prominent over the relevant timescale. Indeed, eyeblink conditioning simulations where slow or fast GCs were removed, the efficiency of generating PC pauses for short and long intervals were reduced (Fig. S2). $CCM_{STP}$ simulations thus demonstrate that a GC temporal basis generated by MF-GC STP is sufficient to reproduce the cerebellar cortex computation underlying delay eyelid conditioning and suggests that the timescale of GC dynamics influences the timescale of behavioral learning.

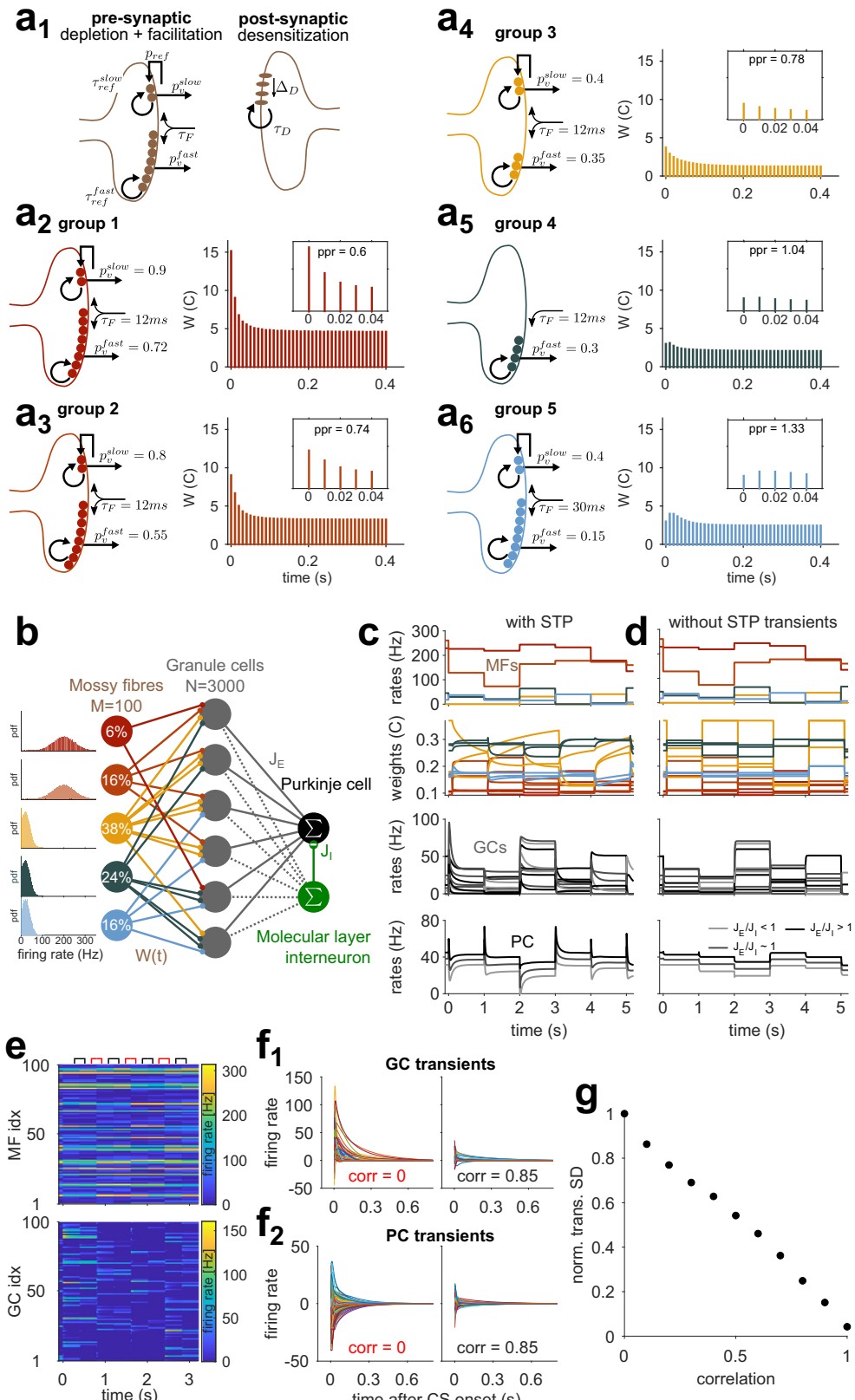

## Analysis of the synaptic mechanism underlying GC transient responses using a reduced model

PC temporal learning requires transient GC activity responses, which in our model only arise from STP at the MF-GC synapse[30,39]. How are the dynamics of synapses and GCs determined by quantal and firing rate parameters? The complexity of the full CCM$_{STP}$ with many interacting

parameters makes it difficult to assess the effect of each synaptic parameter. To overcome this challenge, we developed a reduced MF-GC synapse model, which was analytically solvable for an instantaneous and persistent switch of MF rates. This allowed us to identify the key computational building blocks of CCM$_{STP}$ and explore how they control the overall behavior of the model. Specifically, we omitted

**Fig. 1 | Cerebellar cortex model with short-term synaptic plasticity within the input layer (CCM$_{STP}$). a$_1$** Synaptic model scheme showing the principal parameters. **a$_{2-6}$** Properties of the five model synapse types matching experimental groups from ref. 30. Left: Schemes show differences in presynaptic parameters; the postsynaptic side is identical for all groups. Right: average synaptic weights in response to repetitive 100 Hz stimulation as in ref. 39. Insets: First five responses with paired-pulse ratio (PPR) roughly mimic results from ref. 30. Color code for synapse groups is the same as in ref. 30. **b** Scheme of CCM$_{STP}$. MFs are classified according to the groups in (**a**). Percentages indicate relative frequency of MF groups. Insets: firing rate distributions for different MF groups. **c** Simulation of CCM$_{STP}$ with randomly drawn $J_E$ weights. First panel: 5 sample MFs. Every second, MF activity is re-drawn from distributions in (**b**). Second panel: Normalized weights

of 10 example MF-GC synapses. Third panel: activity of 10 sample GCs. Last panel: PC activity with different shades of gray indicating different E/I ratios onto the PC. **d** Same as **c** but without STP transient dynamics. Low amplitude GC and PC firing rate transients result from 10 ms GC integration time constant. **e** Example simulation in which correlated (black symbols) and uncorrelated (red symbols) MF patterns were presented to the network in alternation. The correlation coefficient for sequential patterns was ≈0.85. Firing rates are color-coded. **f$_1$** Steady-state subtracted GC responses from simulation in (**e**) for uncorrelated (left) and correlated MF pattern switches (right). **f$_2$** Same as (**f$_1$**) but for PC. **g** Normalized standard deviation of PC transient amplitudes for switches between MF patterns of differing levels of correlation.

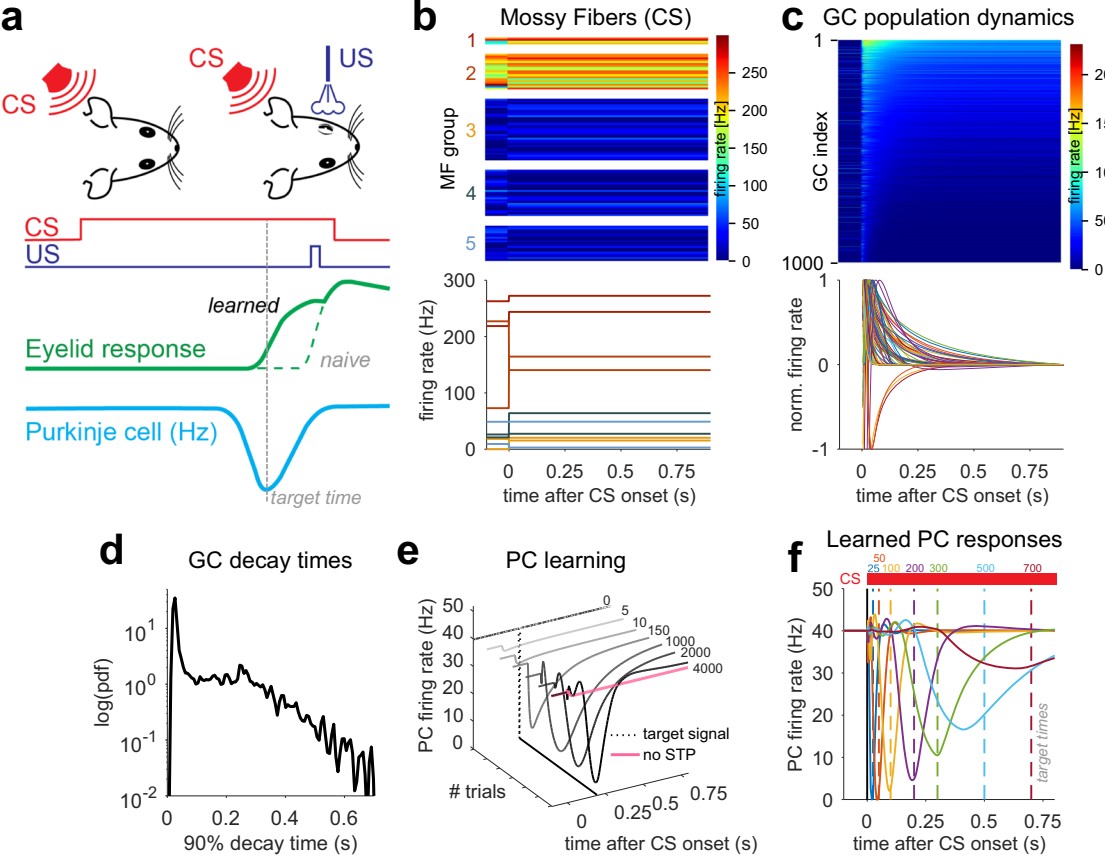

**Fig. 2 | Simulating Purkinje cell pauses during eyelid conditioning. a** Scheme of eyelid conditioning. CS: conditioned stimulus (red). US: unconditioned stimulus (violet). After experiencing CS and US pairings at a fixed temporal interval over many trials, the animal learns to close its eyelid just before the US is delivered (green). A pause in PC activity (blue) precedes the eyelid closure (target time, gray dashed line). **b** The CS is modeled as an instantaneous change in MF firing rate. Top: plot of firing rates of 100 MFs, sorted according to synaptic types (MF groups). MF firing rates are color-coded and drawn according to the distributions shown in Fig. 1b. Bottom: two sample MF rates per synaptic group. Colors as in Fig.1. **c** Model

GC responses to the CS. Top: 1000 GCs sorted according to average firing rate after CS onset. Firing rates are color-coded. Bottom: steady-state subtracted and individually normalized GC transient responses. **d** Pdf of the distribution of GC activity decay times to 10% of the transient peak. **e** Example of delay eyelid conditioning over the course of 4000 learning steps for a 200 ms delay. Dashed line represents the target time used in the supervised learning procedure. Without STP-induced GC transients, no PC pause could be learned (pink line). **f** Simulated PC responses after 4000 learning trials for each target time (colored dashed lines).

short-term facilitation and postsynaptic desensitization and reduced the synaptic model to a single population of high $p_v$ synapses ("drivers"[30]) and a single population of low $p_v$ synapses ("supporters"[30]), each with a fast and a slow refilling ready-releasable pool (Fig. 3b), thus obtaining a model where STP results from vesicle depletion only. Each GC received exactly two driver and two supporter MF inputs with random and pairwise distinct identities (Fig. 3a).

In this reduced model, an instantaneous and persistent switch of MF firing rates generates an average postsynaptic current ($I_{syn}(t)$) for each vesicle pool that is remarkably simple. It features a sharp transient change, followed by a mono-exponential decay to a steady-state

synaptic current amplitude, $A_s$, (Fig. 3c) and can be generally expressed as

$$I_{syn}(t) = A_s + A_t e^{-\frac{t}{\tau_{syn}}} \tag{1}$$

Here, $A_s$ is a time-invariant component and $A_t e^{-\frac{t}{\tau_{syn}}}$ is a transient component with synaptic relaxation time constant $\tau_{syn}$ (Fig. 3c) and amplitude $A_t$. This transient component determines the synapse's ability to encode the passage of time.

The solution of the synaptic dynamics model reveals the crucial dependence of $\tau_{syn}$ and $A_t$ on the presynaptic and firing rate

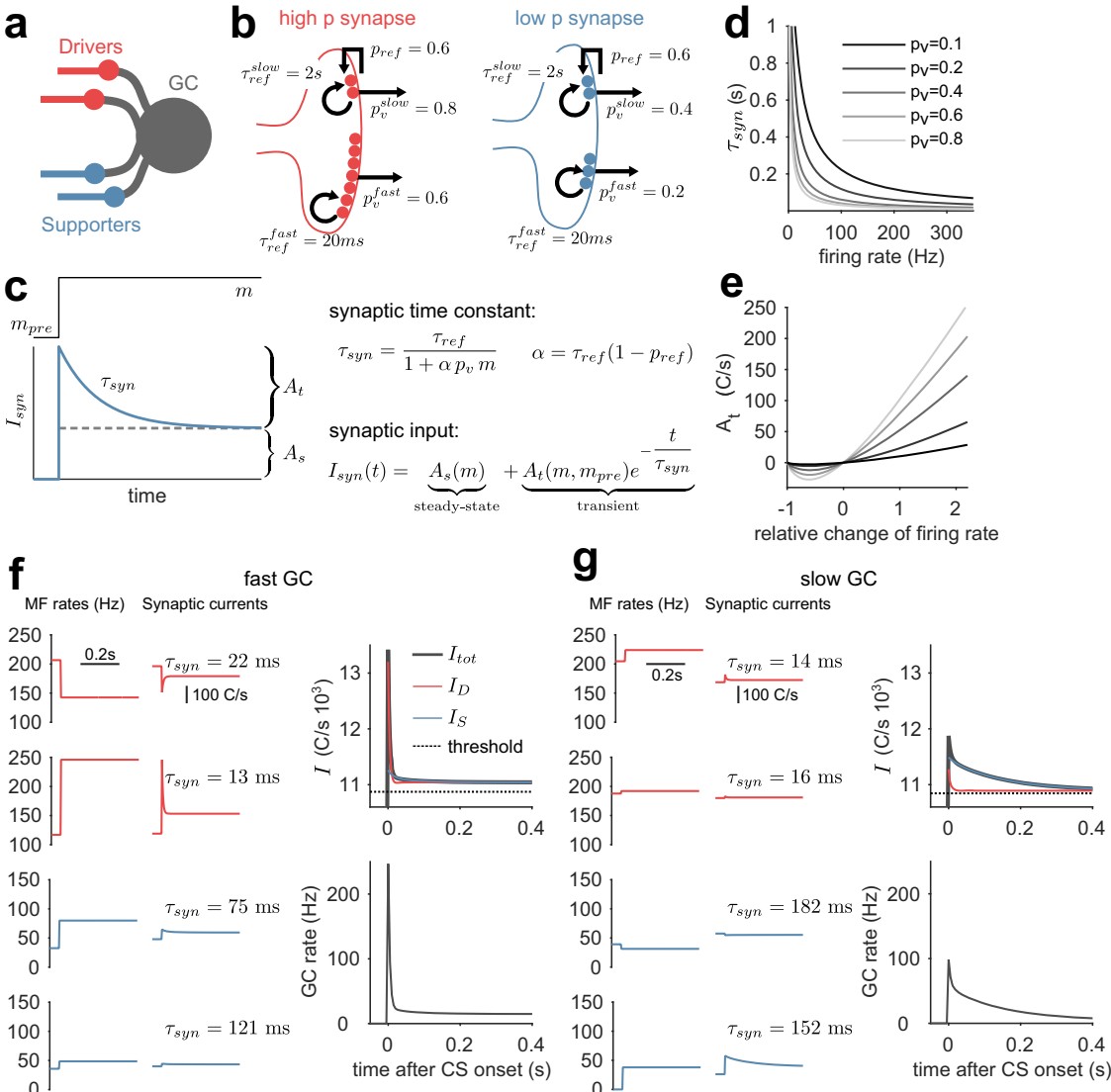

**Fig. 3 | MF-GC synaptic time constants and their relative weights determine the time course of GC responses. a** Scheme of GC inputs in the simplified synaptic model. Each GC receives exactly two distinct high release probability driver (red) and low release probability supporter MFs (blue). **b** Schemes of the reduced synaptic model of high $p_v$ (red) and low $p_v$ synapses (blue). **c** Left: scheme of single pool response with the time constant $\tau_{syn}$ (blue line) to a firing rate switch during CS presentation (black solid line). The dashed black line separates the transient ($A_t$) from the steady-state amplitude ($A_s$). Right: equations determining the synaptic time constant and synaptic input. **d** Slow vesicle pool time constant ($\tau_{syn}$) versus presynaptic MF firing rate. Different shades of gray indicate different release probabilities. **e** Driver synapse transient amplitude ($A_t$) versus relative firing rate

change for a baseline firing rate of 80 Hz (m−$m_{pre}$/$m_{pre}$, $m_{pre}$= 80 Hz). A negative $A_t$ corresponds to a transient decrease in firing rate. Same color code as in (**d**). **f** Sample fast GC. Left: driver and supporter MF firing rates drawn from thresholded normal distributions ($\mathcal{N}_{thr}$(200 Hz, 15 Hz) and $\mathcal{N}_{thr}$(25 Hz, 15 Hz), respectively) and the corresponding synaptic responses. For clarity, only the $\tau_{syn}$ of the respective slow pool is indicated. Upper right panel: GC threshold (dashed line), total synaptic input (black line), total driver input (red line), and total supporter input (blue line). The transient response is dominated by the driver input (red). Lower right panel: resulting GC firing rate response. **g** Like (**f**) but for a sample slow GC. The transient response is dominated by the supporter input (blue).

parameters (see "Methods"):

$$\tau_{syn} = \frac{\tau_{ref}}{1+\alpha p_v m} \qquad (2)$$

Here, $\alpha = \tau_{ref}(1 - p_{ref})$, $m$ is the MF firing rate persisting during the CS, and the synaptic parameters $p_v$, $\tau_{ref}$, and $p_{ref}$ are defined as above. Equation (2) shows that $\tau_{syn}$ is inversely related to the MF firing rate during the CS and the release probability, $p_v$ (Fig. 3d). Intuitively, this is because higher $p_v$ and/or $m$ lead to a higher rate of synaptic vesicle fusion, and hence depletion, driving the synaptic response amplitude to steady-state faster. Conversely, slow time constants arise from low

$p_v$ and/or low $m$ with the maximum $\tau_{syn}$ being equal to the vesicle recovery time $\tau_{ref}$.

The transient amplitude $A_t$ is given by

$$A_t = \frac{N p_v m}{1+\alpha p_v m}\frac{\alpha p_v(m - m_{pre})}{1+\alpha p_v m_{pre}} \qquad (3)$$

Here, $N$ is the number of release sites. Importantly, and in contrast to $\tau_{syn}$, $A_t$ depends on the presynaptic MF firing rate before the CS, $m_{pre}$, and the difference between the MF firing rates before and during the CS. In particular, for both rates sufficiently high, $A_t$ becomes a linear function of the normalized difference between $m$ and $m_{pre}$, i.e.

$A_t \propto (m - m_{pre})/m_{pre}$ (Fig. 3e). $A_t$ is sensitive to the relative and not the absolute change in presynaptic rate, as observed previously[16].

The transient GC activity results from the sum of eight synaptic transient current components, (i.e. four inputs, each with two pools). To illustrate the interplay between the $A_t$ and $\tau_{syn}$, we compared the behavior of each synaptic input for a selected fast and slow GC (Figs. 3f, g). Generally, synaptic inputs from supporters display longer transient currents than synaptic inputs from drivers (Figs. 3f, g, middle panels) due to their lower firing rates (Figs. 3f, g, left panels) and low $p_v$ (Fig. 3b). $A_t$ is largely determined by the relative change in the respective presynaptic MF firing rates, $(m - m_{pre})/m_{pre}$(Fig. 3f and g, left panels). Thus, "fast" GCs are generated when the high $p_v$ driver inputs exhibit large relative changes in firing rates (Fig.3f). "Slow" GCs are generated from synapses with a small relative change in driver firing rates, but large relative supporter (low $p_v$) rate changes paired with low supporter rates during the CS (Fig.3g). Taken together, in the reduced model $\tau_{syn}$ and $A_t$ determine the effective timescales of the GC responses and are explicitly influenced by quantal parameters, synaptic time constants, and the diversity of MF firing rates.

## The explicit influence of synaptic parameters on temporal learning

Our simulations suggest that delay eyelid conditioning across multiple delays necessitates GC population dynamics spanning multiple timescales (Fig. 2, Fig. S2). Since individual GC firing rate dynamics depend on the $A_t$ and $\tau_{syn}$ of their synaptic inputs (Fig. 3), this implies that 1) the spectrum of $\tau_{syn}$ available to the network should cover the relevant timescales and 2) the $A_t$ associated with different $\tau_{syn}$, which can be understood as the relative weights of synaptic transient components, should be of comparable magnitude across $\tau_{syn}$. To illustrate these points, we used the reduced $CCM_{STP}$ to simulate eyelid response learning with different firing rate properties and examined the relationship between $\tau_{syn}, A_t$, the GC temporal basis, and learning outcome. Importantly, since $A_t$ and $\tau_{syn}$ are not independent, the quantity of interest is their joint distribution. We initially set up a reference simulation by choosing MF firing rate distributions such that the diversity of GC transient responses and the temporal learning performance (Fig. 4a) were comparable to the $CCM_{STP}$ with native synapses (Fig. 2f). For this case, the joint distribution shows that $A_t$ decreased with increasing $\tau_{syn}$. Note that $A_t$ is maximal when the MF firing rates increased from zero $m_{pre}$ to a finite $m$ upon CS onset, maximizing $m$-$m_{pre}$ (Eq. 3, see also Fig. S3b, c). We quantified learning accuracy by calculating an error based on 1) the PC response amplitude, 2) its full width at half maximum and 3) the temporal deviation of its minimum from the target delay (Fig. 4a, fifth panel, Fig. S2a, see "Methods""). Importantly, the degradation in temporal precision of the learned PC pauses for longer CS-US intervals was concomitant with the reduction of the $A_t$ associated with longer $\tau_{syn}$ (Fig. 4a). This suggests that inspection of the joint distribution of $\tau_{syn}$ and $A_t$ can provide insight into the temporal learning performance of the network.

When changing only the mean firing rate of *supporter* MFs ($\mu_S$) from 25 Hz to 70 Hz, the synaptic time constants were shortened due to the inverse relationship between $\tau_{syn}$ and the mossy-fiber firing rate $m$ (Fig. 4b, second panel). Consequently, and expectedly, the distribution of GC firing rate decay times was shifted to shorter values, and learning performance was degraded for all CS-US intervals, except the 25 ms delay (Fig. 4b). Lowering the mean firing rate of *driver* MFs ($\mu_D$) from 200 Hz to 100 Hz and increasing the standard deviation ($\sigma_D$) from 15 Hz to 50 Hz, led to an overall increase of the time constants contributed by driver synapses, as well as an increase in their relative weight ($A_t$; Fig. 4c, second panel, marginals). As a result, the joint probability distribution shows a shift towards faster weighted time constants. It also follows that GC transients are accelerated, and learning precision is decreased for long CS-US intervals. Removing synaptic currents originating from driver synapses only disrupted

learning PC pauses for the shortest CS-US interval (Fig. 4d). Reduced model simulations with systematic parameter scans across a wide range of MF firing rate distributions for both synapse types suggested that good synaptic regimes for temporal learning are achieved when driver synaptic weights are comparable or smaller than those of the slow supporting synapses (Fig. S4).

All the results taken together suggest that optimal learning occurs when the spectrum of $\tau_{syn}$ available to the network covers behaviorally relevant timescales with balanced relative weights ($A_t$). Synaptic and GC activity timescales can therefore be tuned by simultaneously modulating $p_v$ and the absolute scale of $m$ to provide the necessary distribution of $\tau_{syn}$, whereas the relative change of MF firing can be used to tune the weight ($A_t$) of $\tau_{syn}$.

## Firing rate and synaptic parameters that improve temporal learning performance

Thus far, we used the reduced model to explore how MF firing rates and synaptic properties influenced the timescales of GC activity and the temporal precision of learned PC pauses. The model, however, was constrained by (1) the use of only two synapse types, (2) fixed release probabilities ($p_v$), (3) MF firing rates that were consistently higher for high $p_v$ synapses than their low $p_v$ counterparts, and (4) an equal number of driver and supporter synapses. We next considered how the relaxation of these assumptions and specific parameter combinations could influence the precision of learned PC pauses. In particular, we simulated reduced models where, in addition to MF firing rates, $p_v$ was sampled from continuous distributions.

Equation (2) suggests that a positive correlation between $p_v$ and $m$ should broaden the distribution of $\tau_{syn}$ and broaden the time window of learning. Specifically, we expect learning performance to improve when high(low) firing rate MFs are, on average, paired with high(low) $p_v$ synapses. We chose uniformly distributed $p_v$ and MF firing rates and split both of these equally into two contiguous groups (Fig. 5a). We performed training simulations in which we paired high $p_v$(driver) synapses with high firing rates, or we paired low $p_v$ (supporter) synapses and high MF firing rates, and vice versa (Fig. 5b). Formally, this is equivalent to adjusting the rank correlation ($c_{rk}$) between the $p_v$ category (supporter or driver) and the $m$ category (high or low, Fig. 5b). We found better learning performance when $p_v$ and $m$ were positively correlated (Fig. 5c, Fig. S5). Indeed, primary vestibular afferents that form driver-like synapses have been shown to have high firing rates[30,40] while supporter-like secondary vestibular afferents have low firing rates[30,41].

Inspired by the number of synapse types observed experimentally[30], we augmented the number of synapse groups from 2 to 5 without changing the $p_v$ and firing rate distributions (Fig. 5d). We reasoned that the introduction of a larger number of MG-GC synapse types would in principle permit a stronger *linear* correlation between $p_v$ and $m$ to occur (Fig. 5e), leading to a broader $\tau_{syn}$ spectrum (not shown) and an improved learning of PC pauses. Indeed, for high $c_{rk}$, the learning performance of the five group $CCM_{STP}$ was better than that of the two-group $CCM_{STP}$ (compare Fig. 5c and Fig. 5f, Fig. S5). These simulation results suggest that good temporal learning performance of $CCM_{STP}$ can be achieved not simply by generating variability in parameters, but by structuring, or tuning, the relationship between $p_v$ and $m$.

Equipped with an understanding of how the synaptic and MF rate parameters can generate different synaptic time constants, we set out to further improve the temporal learning for longer CS-US delays by adjusting the variance of the clustered MF rate distributions. To increase the weighting of long $\tau_{syn}$, we inversely scaled the variance of the MF firing rate distributions with respect to the mean firing rate (Fig. 5g), thereby increasing $A_t$ (Fig. 4c). As expected, PC pause learning was better than when using equal-width MF groups (Fig. 5g, Fig. S5). An additional enhancement of

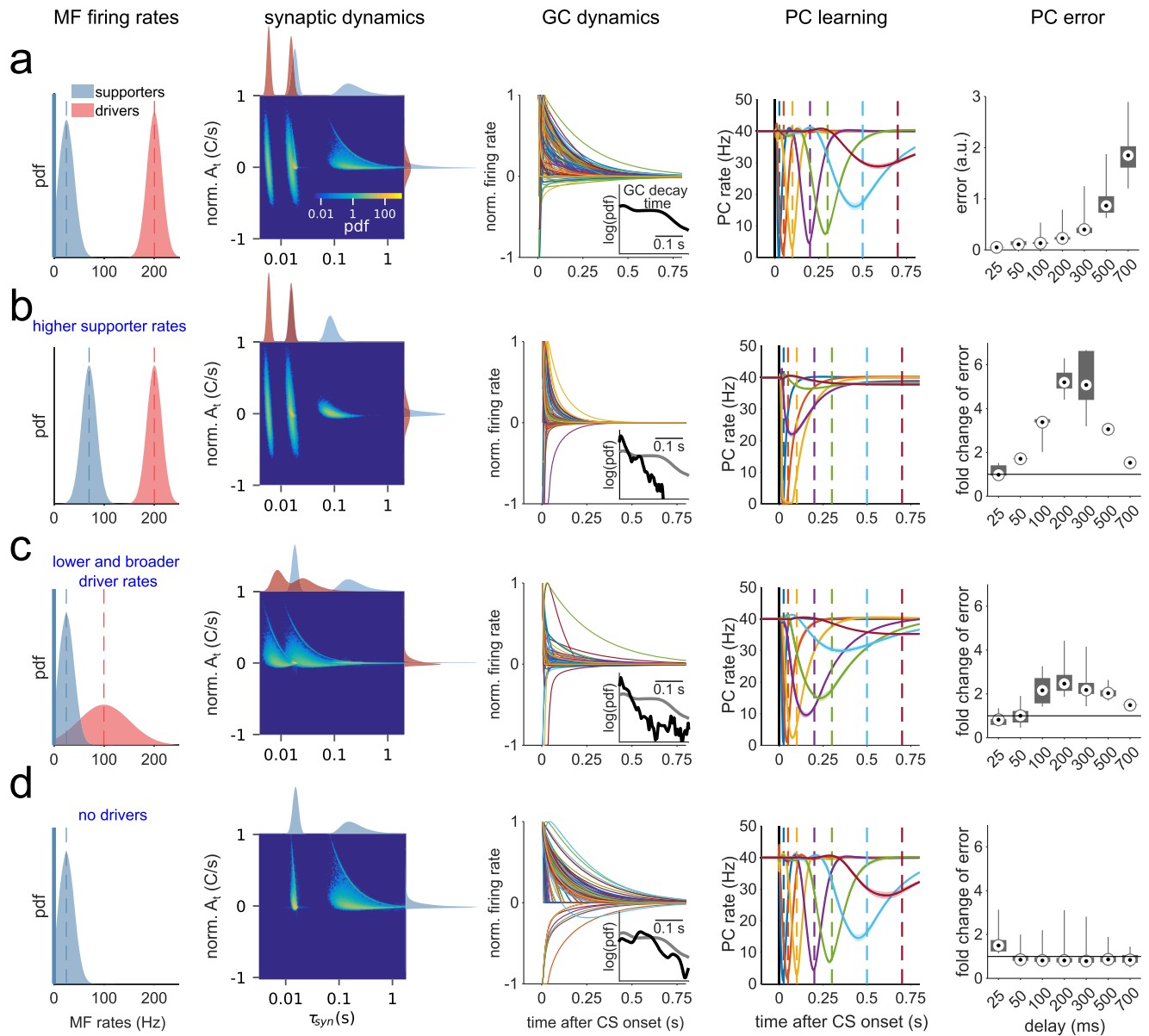

**Fig. 4 | Learning performance depends on MF firing rate distributions. a** First panel: Driver and supporter MF firing rate pdfs ($\mu_D = 200$ Hz, $\mu_S = 25$ Hz, $\sigma_D = \sigma_S = 15$ Hz). Second panel: Resulting joint $A_t$ and $\tau_{syn}$ distribution, featuring four partially overlapping clusters, corresponding to the slow and fast pools for driver and supporter synapses, and marginal distributions. The color code of the joint distribution scales logarithmically. Colors of marginal distributions indicate driver (red) and supporter (blue) components. Third panel: Normalized GC transient responses to CS. Inset: pdf of the distribution of decay times to 10% of the transient peak. Fourth panel: learned PC pauses, averaged over $n = 20$ simulations with different realizations of MF patterns and MF-GC connectivity. Dashed lines mark CS-US intervals (color code is the same as in Fig. 2e). Fifth panel: Error for each CS-US interval is calculated based on PC response amplitude, full-width at half-maximum and temporal deviation (Fig. S2a) and averaged over $n = 20$ realizations of MF patterns and MF-GC connectivity. Black lines indicate the distribution ranges; gray boxes indicate the 25th to 75th percentile range and black-white circles the medians. **b** Same as **a**, but with $\mu_S = 70$ Hz. Inset: black line is the pdf for simulation with $\mu_S = 70$ Hz and gray line is the pdf from (**a**) for comparison. Fifth panel: change in error relative to the average error in (**a**). **c** Same as (**a**), but with $\mu_D = 100$ Hz and $\sigma_D = 50$ Hz. **d** Same as (**a**), but without driver inputs.

learning performance could be achieved by adding a small fraction of zero-rate MFs to the lowest group (Figs. 5g, 6% zero MFs, same fraction as in Fig. 4a), which provide maximal $A_t$ (see Fig. 4). Finally, taking into account the experimental finding that low $p_v$ synapses are more frequent than high $p_v$ synapses[30], we doubled the fraction of MFs and release probabilities in the lowest group, resulting in the best performance of all versions of CCM$_{STP}$ tested here (Fig. 5g). These simulations show that positive correlations between vesicle release probability and presynaptic firing rate broaden the temporal bandwidth of circuit dynamics and improve temporal learning.

## STP permits learning optimal estimates of time intervals

Humans and animals have an unreliable sense of time and their timing behavior exhibits variability that scales linearly with the base interval[47]. Previous work has found that humans seek to optimize their time interval estimates by relying on their prior expectations. A canonical example of this optimization is evident in the so-called ready-set-go-task[48] in which subjects have to measure and subsequently reproduce different time intervals. It has been shown that when the intervals are drawn from a previously learned probability distribution (i.e., prior), subjects integrate their noisy measurements with the prior to generate optimal Bayesian estimates. For example, when the prior distribution is

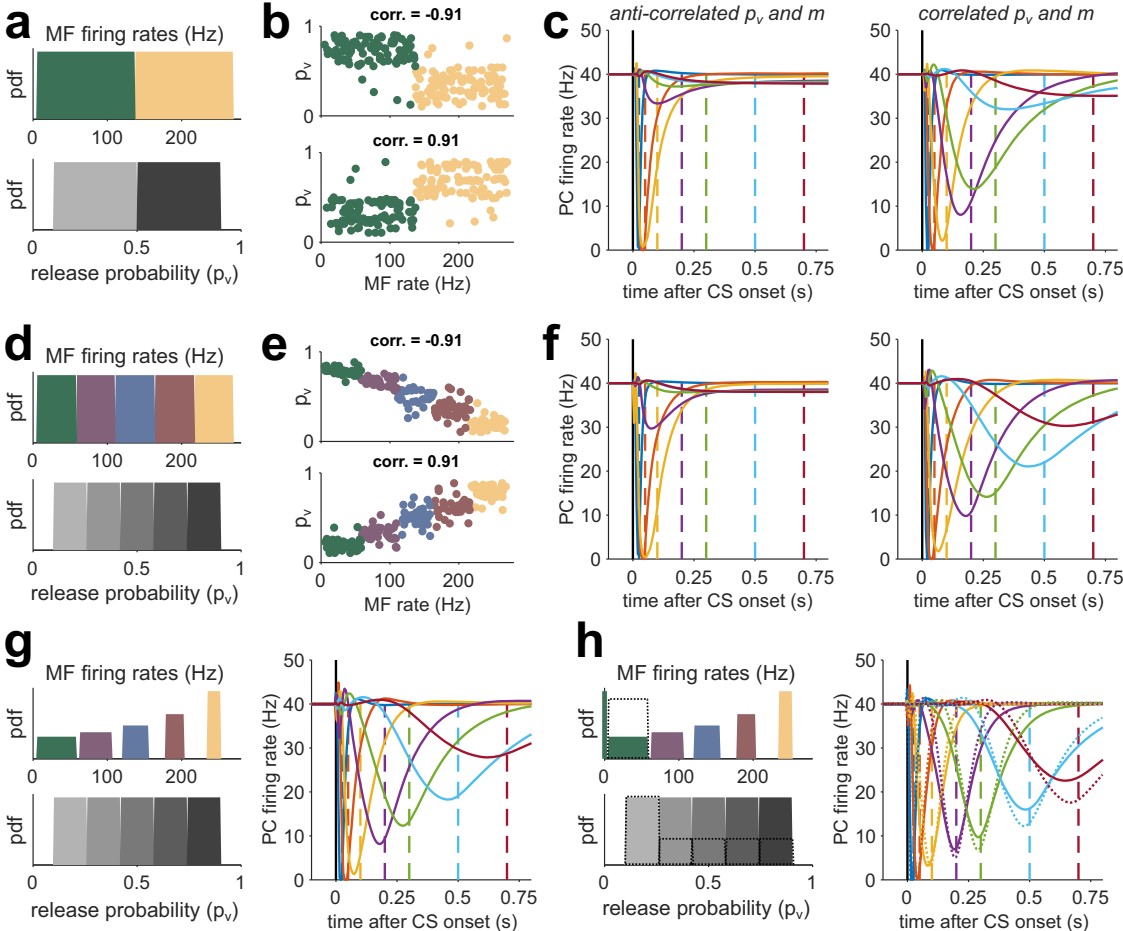

**Fig. 5 | Correlating release probability and MF firing rates improves learning performance. a** Top: distribution of MF firing rates (*m*) used to drive the network, divided into low (supporter, green) and high (driver, yellow) rates. Bottom: Distribution of synaptic release probabilities ($p_v$), divided into low (light gray) and high (dark gray) probabilities. **b** Top: $p_v$ versus m for 500 sample synapses for a network with a strong negative rank correlation between the *m* category (supporter or driver) and the $p_v$ category (high or low). Bottom: same as top, but for strongly positive correlated *m* and $p_v$. **c** Learned PC pauses for low (left) and high (right) correlations. CS-US intervals are color-coded as in Fig. 2f. Each curve is the average

of *n* = 20 simulations with different realizations of MF patterns and MF-GC connectivity. **d–f** Same as (**a–c**), but for distributions divided into five groups. **g** Left: MF rates and release probabilities for five synapse types where the average group firing rate is as in (**d**), but the firing rate variance progressively decreases with the average rate. Right: resulting PC eyelid response learning for high correlations. **h** Same as (**g**) but with zero-rate MFs added to the lowest rate distribution. Dashed lines indicate the case when the count of lowest rates and release probabilities is doubled.

uniform, interval estimates are biased towards the mean of the prior, and biases are generally larger for longer intervals that are associated with more variable measurements (Fig. 6c). Such Bayes-optimal temporal computations are evident in a wide range of timing tasks such as time interval reproduction[48], coincidence detection[49], and cue combination[50].

A recent study developed a cerebellar model called TRACE for temporal Bayesian computations[33]. TRACE implements Bayesian integration by incorporating two features. First, it assumes that GCs form a temporal basis set that exhibits temporal scaling. This feature accounts for the scalar variability of timing. Second, it assumes that prior-dependent learning alters the GC-PC synapses. This feature allows the dentate nucleus neurons (DNs) downstream of PCs to represent a Bayesian estimate of the time interval.

In our analysis of eyelid conditioning (Fig. 2), we showed that CCM$_{STP}$ generates PC firing rate pauses whose width and amplitude linearly scale with time (Fig. 6a). Therefore, we reasoned that CCM$_{STP}$ might have the requisite features for Bayesian integration. To test this possibility quantitatively, we presented our model with variable intervals drawn from various prior distributions. The interval was introduced as a tonic input to MFs, similar to the CS in the eyelid

simulations. The onset of this tonic input caused an abrupt switch of the MF input rates that persisted over the course of a trial. During learning, we subjected the model to intervals sampled randomly from a desired prior distribution.

We tested CCM$_{STP}$ with five different uniform distributions of ready-set intervals (25-150 ms, 50–200 ms, 100–300 ms, 200–400 ms, 300–500 ms), resulting in PC pauses that broadened for longer interval distributions, and integrated DN activity that could easily match the Bayesian least-square model[33] by adjusting a single parameter, the Weber fraction $w_{weber}$ (see "Methods'"; Fig. 6d, h). The reduced model interval estimates were more similar to the Bayesian estimates than for CCM$_{STP}$ with native synaptic parameters, especially for the 200–400 ms and 300–500 ms intervals (Fig. 6h–k). Nevertheless, in both cases the CCM$_{STP}$ simulations show that a GC basis generated by MF-GC STP is sufficient for driving Bayesian-like learning of time intervals spanning several hundreds of milliseconds. It should be noted that our GC temporal basis was not explicitly constructed to accommodate scalar properties. Nevertheless, as in the TRACE model, we observed that interval estimates were biased towards the mean and that these biases were larger for longer intervals. These results suggest that a GC basis set generated from the diverse properties of native

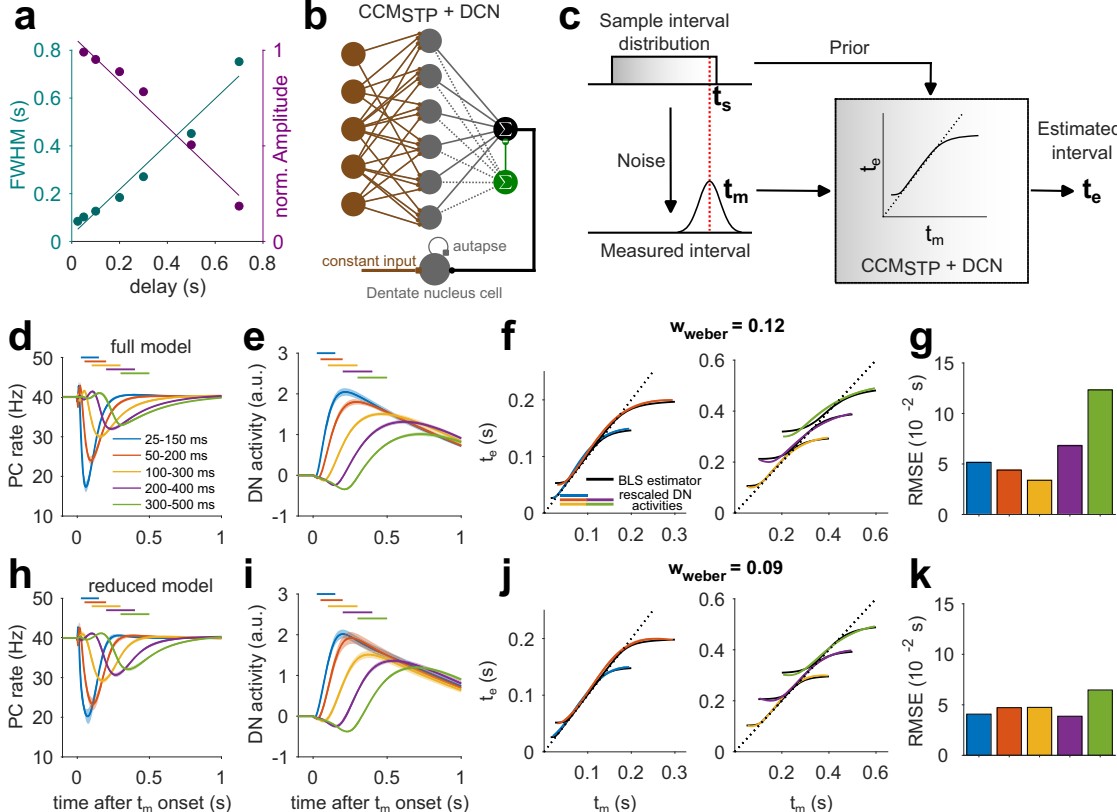

**Fig. 6 | STP-generated temporal basis enables the computation of Bayesian time-interval estimates. a** Full width at half maximum (cyan) and normalized amplitudes (magenta) of learned PC pauses versus delay interval from the experimentally constrained CCM$_{STP}$ (see Fig. 2). Solid lines are linear fits. **b** Scheme of CCM$_{STP}$ with added dentate nucleus cell. **c** Scheme of Bayesian integration. The sample interval $t_s$ (red dashed line, here drawn from a uniform distribution, upper left) is subject to a noisy measurement yielding a measured interval $t_m$ (lower left). CCM$_{STP}$ implements Bayesian integration yielding an estimated interval $t_e$ (right). **d** PC responses after 12,000 learning trials, averaged over $n = 20$ simulations with different realizations of MF patterns and MF-GC connectivity. Shaded area indicates standard deviation. Different colors represent learning of different uniform sample interval distributions. **e** Same as (**d**), but for DN cell activity. **f** Rescaled DN cell activity for different learned interval distributions (colored) and fitted theoretical Bayesian least squares (BLS) estimator (solid black line), with $w_{weber} = 0.12$ resulting from fit. **g** Squared deviation of rescaled DN activity from the BLS estimator for all tested intervals. **h**–**k** Same as (**d**–**g**), but for the reduced model. The reduced model firing rate parameters were $\mu_D = 200$ Hz, $\mu_S = 20$ Hz, $\sigma_D = 10$Hz and $\sigma_S = 15$ Hz and resulted in DN activity consistent with a Bayesian least squares model with $w_{weber} = 0.09$.

---

MF-GC synapses likely exhibits a scalar property necessary for generating optimally timed behaviors.

## Discussion

In order to generate temporally precise behaviors, the brain must establish an internal representation of time. This theoretical study posits that the diversity of synaptic dynamics is a fundamental mechanism for encoding sub-second time in neural circuits. By using eyelid conditioning as a benchmark task for the CCM$_{STP}$, we elucidated the conditions under which the variability in MF-GC synaptic dynamics generates a GC temporal basis set that represents elapsed time and is sufficient for temporal learning on a sub-second scale. According to David Marr's levels of analysis of information processing systems[51], our study connects all three levels, from the circuit computation (learning timed PC pauses) to its underlying algorithm (learning with a temporal basis set), and the fundamental biological mechanism (STP diversity).

### STP diversity as a timer for neural dynamics

Cerebellar adaptive filter models posit that GCs act as a heterogeneous bank of filters that decompose MF activity into various time-varying activity patterns - or temporal basis functions - which are selected and summed by a synaptic learning rule at the PC to produce an output firing pattern that generates behaviors that minimize error signals

arriving via climbing fibers[36,37]. CCM$_{STP}$ can be viewed as an adaptive filter in which MF-GC synapses act as non-linear elements whose filter properties are determined by the experimentally defined synaptic parameters and modulated by the presynaptic MF firing rates.

Recent theoretical work proposes that a scale-invariant neuronal representation of a temporal stimulus sequence can be obtained by using a population of leaky integrators that produce exponentially decaying neural activity transients[52]. Indeed, exponential-like activity has been observed in the entorhinal cortex—a region that projects to the hippocampus[6]. The exponential-like population activity is reminiscent of the GC temporal basis set in CCM$_{STP}$ following persistent firing rate changes. However, the MF-GC synaptic inputs are always a mixture of multiple exponential components. Nevertheless, our work suggests that STP could be a plausible biological mechanism explaining exponential dynamics in neuronal populations[6] and merits further theoretical and experimental investigation.

The use of an instantaneous and persistent change in MF activity was motivated by the fact that eyelid conditioning can be achieved if the CS is replaced with a constant MF stimulation[44,45,53]. Recent evidence from pons recordings during reaching suggests that MF activity can be persistent with little dynamics[54]. For dynamic changes in MF rates, STP is likely to generate outputs that are phase-shifted and/or the derivatives of their input[55]. Using heterogeneity of MF-GC STP as a mechanism for adaptive filtering, even time-varying inputs will

effectively be diversified within the GC layer and improve the precision of temporal learning.

Synapses within the prefrontal cortex[56] and at thalamocortical connections[57] exhibit diverse firing rate inputs and release probabilities[58], generating synaptic dynamics that could drive complex neural dynamics. Reminiscent of PC firing rate pauses during eyelid conditioning, hippocampal time cells are thought to be generated by a linear combination of exponentially decaying input activity patterns from upstream entorhinal cortical neurons[6]. More generally, it has been shown that STP also provides a critical timing mechanism within a recurrent neural network model of neocortical activity by facilitating temporal pattern descrimination[18]. We note that all synapses in this study featured only a single STP timescale, but we expect that the addition of heterogeneous STP would further diversify the network's dynamics and enhance its computational properties. Thus, these previous studies and our present study underscore the proposal that STP diversity is a tunable timing mechanism for generating neural dynamics across brain regions.

### Timing mechanisms in the cerebellar cortex

In addition to MF-GC STP, the cerebellar cortex is equipped with multiple mechanisms potentially enabling temporal learning[59]. Indeed, time-varying MF inputs could directly provide a substrate for learning elapsed time[60], but whether the observed diversity of MF firing is sufficient to mediate temporally precise learning is unknown and merits further exploration. Within the cerebellar cortex, unipolar brush cells are thought to provide delay lines to diversify GC activity patterns[10,61,62], but these cell types are rare outside the mammalian vestibular cerebellum. The diversity of GC STP[63] could add to the diversity of the effective GC-layer basis set[64]. Consistent with the importance of MF-GC STP, delay eyelid conditioning was selectively altered due to the loss of fast EPSCs in AMPAR KO mice[65]. Simulations including realistic NMDA and spillover dynamics[66] can further enrich the temporal scales available to the network[67]. It would be of particular interest to investigate the role of MF-GC STP in the context of recurrent GC-Golgi-Cell-cell network models that have been shown to generate rich GC temporal basis sets[12,29]. Finally, we note that MF-GC STP and other timing mechanisms described above are not mutually exclusive but presumably act in concert with the diverse intrinsic properties of GCs[68] and PCs[69] to cover different timescales of learning or increase mechanistic redundancy.

### Predictions of the CCM$_{STP}$

Our theory makes several testable predictions. The transient response amplitude of PCs, which is proportional to the relative change in firing rate, can serve as a detector of rapid changes in MF firing patterns (novelty) and thus amplify pattern discrimination similar to that demonstrated for synapse-dependent delay coding[30]. Consistent with this prediction, single whisker deflections have been shown to generate transient PC activity[42].

CCM$_{STP}$ predicts that persistent changes in MF activity would generate exponential-like GC activity profiles (Figs. 2, 4). However, although the majority of simulated GCs shown here are active at the onset of the CS, this is not a necessary feature of CCM$_{STP}$. When we included a single, average-subtracting Golgi cell (possibly representing the "common mode" of Golgi Cell population activity[64]), more GCs showed delayed onset firing and the variability of onset and peak times (Fig. S6). This did not affect the learning performance of simulated delay eyelid conditioning (Fig. S6). Note that our implementation of Golgi cell feedback is simplified and does not account for reciprocal inhibition between multiple Golgi cells, which in simulations has also been shown to generate diverse GC activity[12,29]. To test these predictions, MFs could be driven at constant rates using direct electrical or optogenetic stimulation of the cerebellar peduncle in vivo or the white matter in acute brain slices, with and without intact Golgi cell

inhibition. Unfortunately, high-temporal resolution population recordings of GCs are challenging due to the small size of GC somata. In the future, small impendence silicon probe recordings[70] or ultra-fast optical indicators[71] might permit experimentally testing our hypotheses. If successful, we predict that the time course of GC responses should be diverse and exponential-like, with prominent delayed activity in some granule cells when Golgi cells are intact. Furthermore, decreasing or increasing the MF firing rate should in turn slow or accelerate GC responses, respectively. Finally, for complex behavioral experiments in which the MF activity is dynamic (and measurable), one could examine which circuit connectivity of the CCM$_{STP}$ best reproduces the measured GC activity.

The CCM$_{STP}$ is one of the few network models directly linking quantal synaptic parameters and presynaptic activity dynamics to population activity dynamics and temporal learning. Figures 3 and 4 show that the relative weight and temporal span of synaptic time constants dictate the distribution of GC firing rate decay times and, in turn, the timescales of temporal learning. Analytical solutions for simple synapse models (Eq. (3)) provide insight into how synaptic parameters influence STP. For example, high levels of correlation between $p_v$ and $m$, coupled with balanced relative weights of the synaptic time constants, generated a learning performance superior to the native synapses (Fig. 5d). Therefore, CCM$_{STP}$ predicts that MFs forming driver synapses (high $p_v$) would have a high baseline and stimulated firing rates, while MFs forming supporter synapses (low $p_v$) would exhibit low baseline and stimulated firing rates, albeit with large relative changes in firing rates. Indeed, vestibular neurons, which have been shown to exhibit high firing rates[72,73], produced MF-GC synapses with high release probability[30]. In the C3 zone of the anterior lobe in cats, specific firing rates were associated with different MF types[74]. It is tempting to hypothesize that nature tunes presynaptic activity and synaptic dynamics (perhaps by homeostatic or activity-dependent mechanisms) in order to preconfigure the window of temporal associations required for a particular behavior.

### Choice of the cerebellar learning rule

The learning rule we used here was adapted from a previous modeling study that investigated cerebellar adaptation of the vestibular ocular reflex and was argued to be biologically plausible[75]. This synaptic weight update rule is mathematically equivalent to a gradient descent in which the error magnitude is transmitted via the climbing fiber[75]. Consequently, CCM$_{STP}$ learning rule features graded climbing-fiber responses and a gradual reduction in climbing-fiber spiking that is concomitant with the progression of learning. These phenomena have been observed experimentally[43,76]. Moreover, a recent study that thoroughly investigated the role of the climbing fiber spike in cerebellar learning found that the GC and climbing-fiber spike pairings necessary for the induction of long-term depression/potentiation under physiological conditions are compatible with a stochastic gradient descent rule[46]. The CCM$_{STP}$ learning rule can be seen as a deterministic variant of this.

### Synaptic implementation of a Bayesian computation

Bayesian theories of behavior provide an attractive framework for understanding how organisms, including humans, optimize time perception and precise actions despite the cumulative uncertainty in sensory stimuli, neural representations, and generation of actions[48,77]. We found that CCM$_{STP}$ could generate biased time estimates consistent with Bayesian computations. In general, the magnitude of biases for a Bayesian agent depends on the magnitude of timing variability (i.e., Weber fraction). In our simulations, model parameters corresponding to native synapses from the vestibular cerebellum produced biases that were optimal for a typical weber fraction of 0.12. However, CCM$_{STP}$ is flexible and can be adjusted to generate optimal biases for a wide range of weber fractions. The exact relationship

between model parameters and $w_{weber}$ is an important question for future research. We note that the timescales of synaptic properties observed empirically in the vestibular cerebellum[30] are only suitable for generating optimal estimates for relatively short time intervals. Therefore, whether the synaptic mechanisms that underlie CCM$_{STP}$ could accommodate timing behavior for longer timescales remains to be seen. One intriguing hypothesis is that synaptic parameters in different cerebellar regions are tuned to generate optimal estimates for different time intervals, similar to the timing variability observed for cerebellar long-term synaptic plasticity rules[78].

## Methods

### MF-GC synapse model

The synaptic weight between the $j$th MF and the $i$th GC is denoted by $W_{ij}$. The firing rate of the $j$th MF is represented by $m_j(t)$ and the average current per unit time transmitted by the synapse between GC $i$ and MF $j$ is

$$I_{syn,ij}(t) = W_{ij}(t) \cdot m_j(t). \tag{1}$$

Time-dependent MF-GC synaptic weights were modeled using two ready-releasable vesicle pools[38], each according to the general form established by Tsodyks and Markram[79]. A similar model was shown to accurately describe STP at the MF-GC synapse[38]. Accordingly, one vesicle pool was comparatively small, with a high release probability and a low rate of recovery from vesicle depletion (0.5 s⁻¹), while the other was comparatively large, with low release probability and a high rate of recovery from depletion (20ms⁻¹)[38]. We refer to these pools as 'slow' and 'fast', respectively. In the Hallermann model[38], the slow pool is refilled by vesicles from the fast pool. For the sake of mathematical tractability, we modeled the pools as being refilled independently (see scheme in Fig. 1).

To model vesicle depletion, we use the variables $x^{slow}$ and $x^{fast}$, denoting the fraction of neurotransmitter available at the slow and fast vesicle pool. The state of the pools between GC $i$ and MF $j$ at time $t$ is then described by

$$\dot{x}_{ij}^{slow}(t) = \frac{1-x_{ij}^{slow}(t)}{\tau_{ref}^{slow}} - u_{ij}^{slow}(t) \cdot (1-p_{ref}) \cdot x_{ij}^{slow}(t) \cdot m_j(t)$$
$$\dot{x}_{ij}^{fast}(t) = \frac{1-x_{ij}^{fast}(t)}{\tau_{ref}^{fast}} - u_{ij}^{fast}(t) \cdot x_{ij}^{fast}(t) \cdot m_j(t), \tag{2}$$

where, $\tau_{ref}^{slow}$ and $\tau_{ref}^{fast}$ are the time constants of recovery from vesicle depletion for the slow and fast pools, and are identical for all synapses. The variables $u_{ij}^{slow}(t)$ and $u_{ij}^{fast}(t)$ denote the pools' respective release probabilities at time $t$. Experimental data show that, in response to trains of action potentials, MF-GC synapses approach synaptic steady-state transmission with a long time constant[38,39]. This feature can be captured with a serial pool model[38] (see scheme in Fig. S7). In order to capture this behavior with a parallel pool model, we added the phenomenological parameter $p_{ref}$ to the slow pool's dynamical equation. In mechanistic terms, $p_{ref}$ can be thought of as the probability of immediately refilling a synaptic docking site after the release of a vesicle. This mechanism effectively mimics a simplified form of activity-dependent recovery from depression. The final release probabilities $u_{ij}^{slow}(t)$ and $u_{ij}^{fast}(t)$ are modulated by synaptic facilitation according to

$$\dot{u}_{ij}^{slow}(t) = \frac{p_{v,slow}^\alpha - u_{ij}^{slow}(t)}{\tau_F^\alpha} + p_{v,slow}^\alpha \cdot (1-u_{ij}^{slow}(t)) \cdot m_j(t)$$
$$\dot{u}_{ij}^{fast}(t) = \frac{p_{v,fast}^\alpha - u_{ij}^{fast}(t)}{\tau_F^\alpha} + p_{v,fast}^\alpha \cdot (1-u_{ij}^{fast}(t)) \cdot m_j(t). \tag{3}$$

Here, $p_{v,fast}^\alpha$ and $p_{v,slow}^\alpha$ denote the release probabilities for the fast and slow pools, respectively, and $\tau_F^\alpha$ is the facilitation time constant.

The index $\alpha$ denotes different synapse types (groups from Chabrol et al.[30]) and varies from 1 to 5. The average number of vesicles released at any time t can be written as:

$$n_{ij}^{slow}(t) = N_{slow}^\alpha \cdot u_{ij}^{slow}(t) \cdot x_{ij}^{slow}(t)$$
$$n_{ij}^{fast}(t) = N_{fast}^\alpha \cdot u_{ij}^{fast}(t) \cdot x_{ij}^{fast}(t). \tag{4}$$

Postsynaptic receptor desensitization induces an additional component of depression of phasic MF-GC synaptic transmission. As both pools share the same postsynaptic target, we model desensitization via the modulation of a single variable $q_{ij}(t)$ for each synapse type, which represents the synaptic quantal size and which is influenced by the total number of vesicles released from both pools:

$$\dot{q}_{ij}(t) = \frac{q_0 - q_{ij}(t)}{\tau_D} - \Delta_D \cdot q_{ij}(t) \cdot \frac{n_{ij}^{slow}(t) + n_{ij}^{fast}(t)}{N_{tot}} \cdot m_j(t) \tag{5}$$

where $N_{tot}^\alpha = N_{slow}^\alpha + N_{fast}^\alpha$, $\tau_D$ is the time constant of recovery from desensitization, $q_0$ is the quantal size in the absence of ongoing stimulation and $\Delta_D$ is a proportionality factor that determines the fractional reduction of $q_{ij}(t)$. As explained below, we set $q_0 = 1$, i.e. $q_{ij}(t)$ is normalized. Both $\tau_D$ and $\Delta_D$ are identical across all synapse types. Finally, the total synaptic weight is equal to the sum of the contributions from both vesicle pools:

$$W_{ij}(t) = q_{ij}(t) \cdot \left( n_{ij}^{slow}(t) + n_{ij}^{fast}(t) \right), \tag{6}$$

### Synaptic parameters for generating diverse synaptic strength and dynamics

We set the synaptic parameters of our model to reproduce the average behavior of the 5 MF-GC synapse groups which were determined in ref. 30 based on unitary response current amplitudes, pair pulse ratios, and response coefficients of variation.

The vesicle pool refilling time constants $\tau_{ref}^{slow}$ and $\tau_{ref}^{fast}$ were set to the values measured at the MF-GC synapse in ref. 38 and were identical for all synapse groups. The time constant of facilitation $\tau_F^\alpha$ for groups 1–4 was taken from ref. 39. The time constant of recovery from desensitization, $\tau_D$, was set equal to the value reported in ref. 38 for all groups, and the parameters $\Delta_D$ was chosen so as to obtain the relative reduction in quantal size reported in the same ref. 38. To qualitatively account for the slow approach to steady-state transmission observed in MF-GC synapses[38,39] we set $p_{ref}$ to a value of 0.6 for all synapse types.

To set the presynaptic quantal parameters, we matched model quantal parameters, $q_O$, $N$ and $p_v$, to the average of those measured in ref. 30 for each synapse group. The estimation of the experimental values $q_{0,exp}^\alpha$, $N_{exp}^\alpha$ and $p_{v,exp}^\alpha$ was carried out via multiple-probability fluctuation analysis[30], which assumes a single vesicle pool. To constrain the corresponding parameters of our two-pool model, we assumed:

$$N_{exp}^\alpha = N_{tot}^\alpha = N_{slow}^\alpha + N_{fast}^\alpha$$

$$p_{v,exp}^\alpha = \frac{N_{slow}^\alpha p_{v,slow}^\alpha + N_{fast}^\alpha p_{v,fast}^\alpha}{N_{tot}^\alpha} \tag{7}$$

while keeping $p_{v,slow}^\alpha > p_{v,fast}^\alpha$. Since the quantal size did not significantly differ between groups[30], we set $q_0 = 1$ for all groups for simplicity. As group 4 featured almost no STP, we modeled these synapses without slow pool.

The above equations do not have a unique solution. In order to constrain the synaptic parameters further, we additionally required that the relative unitary response current amplitudes between synapse groups and their pair pulse ratios approximately equal the

**Table 1 | Synaptic parameters used in full model**

|  | Group 1 | Group 2 | Group 3 | Group 4 | Group 5 | Ref. |
|---|---|---|---|---|---|---|
| $N_{slow}$ | 4 | 3 | 4 | – | 3 | [30] |
| $N_{fast}$ | 16 | 12 | 6 | 10 | 12 | [30] |
| $p_{v,slow}$ | 0.9 | 0.8 | 0.4 | – | 0.4 | [30] |
| $p_{v,fast}$ | 0.72 | 0.55 | 0.35 | 0.3 | 0.15 | [30] |
| $\tau_{ref}^{slow}$ [ms] | 2000 | 2000 | 2000 | – | 2000 | [38] |
| $\tau_{ref}^{fast}$ [ms] | 20 | 20 | 20 | 20 | 20 | [38] |
| $\tau_F$ [ms] | 12 | 12 | – | 12 | 30 | [30,39] |
| $p_{ref}$ | 0.6 | 0.6 | 0.6 | – | 0.6 | – |
| $\Delta_D$ | 0.1 | 0.1 | 0.1 | 0.1 | 0.1 | [38] |
| $\tau_D$ [ms] | 100 | 100 | 100 | 100 | 100 | [38] |
| occurrence | 6% | 16% | 38% | 24% | 16% | [30] |

**Table 2 | MF firing rate parameters used in the full model**

|  | Group 1 | Group 2 | Group 3 | Group 4 | Group 5 |
|---|---|---|---|---|---|
| $\mu$ [Hz] | 200 | 200 | 20 | 20 | 20 |
| $\sigma$ [Hz] | 20 | 20 | 20 | 20 | 20 |

experimentally measured ones. To account for the fact that group 5's pair pulse ratio is larger than one, we set $\tau_F = 30$ ms for this group, as in ref. 30.

Finally, we extracted the relative occurrence of each synapse type from ref. 30.

A set of synaptic parameters that reproduces the behavior of the five synapse groups from ref. 30 that we used in Figs. 1, 2, and 6 is summarized in Table 1.

**MF firing rate parameters**

MF firing rate distributions of the full CCM$_{STP}$ were set according to the broad range described in the literature[40,41,70,72,73,80–84]. MFs forming synapse types 1 and 2, which convey primary sensory information, were set to high firing frequencies according to experimental observations[40,41] (see Fig. 1b, left panels). In contrast, the firing rates for the other synapses types were lower[70,83]. For the full model, this led to synapses with high $p_v$ being associated with MF inputs with comparatively higher average firing rates (primary sensory groups 1, 2) and synapses with low $p_v$ being associated with MF inputs with comparatively lower average firing rates (secondary/processed sensory groups 3, 4, 5). We chose to describe MF firing rate distributions by Gaussian distributions whose negative tails were set to zero. Means and standard deviations of the Gaussian distributions were set such that the means and standard deviations of the resulting thresholded distributions resulted in the values summarized in Table 2.

**Cerebellar cortical circuit model**

The standard cerebellar cortex model with STP (CCM$_{STP}$) consists of firing rate units corresponding to 100 MFs, 3000 GCs, a single PC, and a single molecular layer interneuron (MLI). The PC linearly sums excitatory inputs from GCs and inhibition from the MLI. Each GC receives four MF synapses, randomly selected from the different synapse types according to their experimentally characterized frequency of occurrence[30]. The synaptic inputs to the GCs and their firing rates are given by:

$$I_{gc,i}(t) = \sum_{j\in K} I_{syn,ij}(t) = \sum_{j\in K} W_{ij}(t)m_j(t)$$
$$\tau_g \dot{gc}_i(t) = -gc_i(t) + \alpha_i \cdot \max(I_{gc,i}(t) - \theta_i, 0) \tag{8}$$

where the granule cell membrane time constant $\tau_g = 10$ ms. In the above equation, $K$ is a set of four indices, randomly drawn from all MF. We require that at least one MF per GC belongs to groups 1, 2 or 5, as observed experimentally[30]. The gain $\alpha_i$ and threshold $\theta_i$ are set individually for each GC $i$ as explained below.

MLI activity is assumed to represent the average rate of the GC population, thus allowing each GC to have a net excitatory or inhibitory effect depending on the difference between the MLI-PC inhibitory weight and the respective GC-PC excitatory weight:

$$mli(t) = \frac{1}{N}\sum_{i=1}^{N} gc_i(t), \tag{9}$$

The synaptic weights between the $i$th GC and the PC and between the MLI and PC were defined as $J_{E,i}$ and $J_I$, respectively. The total synaptic input to the PC is thus given by

$$I_{pc}(t) = \sum_{i=1}^{N}\frac{J_{E,i}}{N} gc_i(t) - J_I mli(t) + I_{spont}$$
$$= \frac{1}{N}\sum_{i=1}^{N}(J_{E,i} - J_I)gc_i(t) + I_{spont}. \tag{10}$$

$I_{spont}$ is an input that maintains the spontaneous firing of the PC at 40 Hz.

Finally, the PC firing rate is given by

$$pc(t) = \max(I_{pc}(t), 0). \tag{11}$$

In Fig. 1, the GC-PC weights $J_{E,i}$ were drawn from an exponential distribution with mean equal to 1. To decrease or increase the ratio of the average excitatory to inhibitory weight, in Figs. 1c and 1d we set $J_I = 1.025$ and $J_I = 0.975$, respectively. The full CC model and the reduced model (described below) were numerically integrated using the Euler method with step size 0.5 ms.

**GC Threshold and gain adjustment.** Changing the statistics of the MF firing rate distributions changes the fraction of active GCs at any given time and the average GC firing rates. To avoid the confounding impact that co-varying these quantities has on learning performance when comparing different MF parameter sets, we adjusted GC thresholds, $\theta_i$ and gains $\alpha_i$ such that, at steady state, the fraction of active GCs and the average GC firing rates were identical for all MF parameter choices. Specifically, we drew 1000 random MF patterns from the respective firing rate distributions, and we calculated the steady inputs values of the synaptic dynamics as follows:

$$\left(u_{ij}^{slow,\mu}\right)^* = p_{v,slow}^{\alpha} \cdot \frac{1 + \tau_F^{\alpha}\cdot m^{\mu}}{1 + p_{v,slow}^{\alpha}\cdot \tau_F^{\alpha}\cdot m_j^{\mu}}$$
$$\left(u_{ij}^{fast,\mu}\right)^* = p_{v,fast}^{\alpha} \cdot \frac{1 + \tau_F^{\alpha}\cdot m^{\mu}}{1 + p_{v,fast}^{\alpha}\cdot \tau_F^{\alpha}\cdot m_j^{\mu}} \tag{12}$$

$$\left(x_{ij}^{slow,\mu}\right)^* = \frac{1}{1 + \left(u_{ij}^{slow,\mu}\right)^* \cdot \tau_{ref}^{slow}\cdot (1 - p_{ref})\cdot m_j^{\mu}}$$
$$\left(x_{ij}^{fast,\mu}\right)^* = \frac{1}{1 + \left(u_{ij}^{fast,\mu}\right)^* \cdot \tau_{ref}^{fast}\cdot m_j^{\mu}} \tag{13}$$

$$\left(q_{ij}^{\mu}\right)^* = \frac{N_{tot}}{N_{tot} + \Delta_D\cdot\tau_D\cdot\left(\left(n_{ij}^{slow,\mu}\right)^* + \left(n_{ij}^{fast,\mu}\right)^*\right)\cdot m_j^{\mu}} \tag{14}$$

With these, we obtained, for each GC, the distribution of steady-state inputs and firing rates:

$$\left(I_{gc,i}^{\mu}\right)^* = \sum_{j \in K} \left(W^{\mu}\right)_{ij} m_j^{\mu}(t)$$

$$\left(gc_i^{\mu}\right)^* = \alpha_i \cdot \max\left(\left(\left(I_{gc,i}^{\mu}\right)^* - \theta_i, 0\right)\right) \quad (15)$$

We then adjusted $\alpha_i$ and $\theta_i$ for each GC to maintain an average steady-state GC firing rate of 5 Hz for all patterns. The lifetime sparsity of each GC was set to 0.2, which is within the range of experimental observations[84,85]. Throughout the article, this adjustment was carried out every time we changed synaptic parameters (Fig. 5), the parameters of the MF firing rate distributions (Fig. 4) or the MF to synapse connectivity (Fig. 5).

**Supervised learning rule.** Purkinje cell pauses associated with eyelid conditioning acquisition were generated by adjusting $J_{E,i}$ using a supervised learning rule. The target PC firing rate $I_{target}(t)$ was set as a Dirac pulse in which the PC rate is zero in the time bin around $t_{target}$ following the start of the CS.:

$$I_{target}(t) = I_{spont} \cdot \left[1 - S\left(t - t_{target}\right)\right] \quad (16)$$

where $S = 1$ in the time bin around $t_{target}$ and $S = 0$ otherwise. We quantify the deviation of the PC firing rate from the target rate by the least squares loss $E$ that is to be minimized during learning:

$$E = \frac{1}{2} \int_{-T_{pre}}^{T_{CS}} dt\, \widetilde{w}_{err}^2(t) \epsilon^2(t)$$

$$= \frac{1}{2} \int_{-T_{pre}}^{T_{CS}} dt\, \widetilde{w}_{err}^2(t) \left(I_{pc}(t) - I_{target}(t)\right)^2 \quad (17)$$

$[0, T_{CS}]$ is the time interval after CS onset (at $t = 0$) during which we require the PC to follow the target signal and $[-T_{pre}, 0]$ is a time interval before CS onset during which the PC should fire at its spontaneous rate. $\epsilon(t)$ denotes the deviation between the target and the actual PC output at time $t$. $\widetilde{w}_{err}$ is a factor that we use to increase the sensitivity of the loss E function to the target time, and is given by:

$$\widetilde{w}_{err}(t) = \frac{w_{err}(t)}{\int_{-T_{pre}}^{T_{CS}} dt' w_{err}(t')}$$

$$w_{err}(t) = \begin{cases} 3.5 & \text{if } t = t_{target} \\ 1 & \text{else} \end{cases} \quad (18)$$

In all main figures, we used $T_{CS} = 1.4 s$ and $T_{pre} = 0.1 s$.

GC-PC weights $J_{E,i}$ were modified during learning using gradient descent to reduce the error $E$ at each step of the learning algorithm:

$$J_i \leftarrow J_i + \Delta J_i$$

$$\Delta J_i = \eta \frac{\partial E}{\partial J_i}$$

$$= \frac{\eta}{N} \int_{-T_{pre}}^{T_{CS}} dt\, \widetilde{w}_{err}^2(t) \cdot \epsilon(t) \cdot gc_i(t) \quad (19)$$

Here, $\eta$ is a learning rate. For our simulations, we modified this basic rule in two ways. Firstly, similar to ref. 75, we explicitly simulated a climbing fiber (CF) rate, $cf$, that is modulated by the error signal $\epsilon(t) = I_{pc}(t) - I_{target}(t)$ according to

$$cf(t) = \max(cf_{spont} + \beta\epsilon(t), 0) \quad (20)$$

where $cf_{spont}$ is the spontaneous CF rate and $\beta$ a proportionality factor. The CF rate was then used to update the synaptic weight according to the following equation:

$$\Delta J_i = \frac{\eta}{N} \int_{-T_{pre}}^{T_{CS}} dt\, \widetilde{w}_{err}^2(t) \cdot (cf_{spont} - cf(t)) \cdot gc_i(t) \quad (21)$$

where we also set $J_{E,i} = 0$ when a learning iteration resulted in a negative weight. As the CF rate is required to be positive or zero, this formulation limits the error information transmitted to the PC compared to the simple gradient rule. This learning rule yields synaptic long-term depression when CF and GC are simultaneously active and long-term potentiation when GCs are active alone, consistent with experimental data on GC-PC synaptic plasticity[59].

Furthermore, recent experimental findings suggest that the temporal properties of GC-PC plasticity rules are tuned to compensate for the typical delays expected for error information arriving in the cerebellar cortex[78]. Here, we did not explicitly model CF error information delays, and for the sake of simplicity, directly modeled the timing of PC activity to show that the GC basis set is sufficient to generate an appropriately timed PC pause.

To increase the learning speed, we added a Nesterov acceleration scheme to Eq. (21)[86], introducing a momentum term to the gradient, i.e. weight updates made during a given iteration of the algorithm depended on the previous iteration. The implementation we chose additionally features an adaptive reset of the momentum term, improving convergence properties[86]. This addition is for practical convenience and does not reflect biological mechanisms.

For the weight learning, we subsampled the simulated GC rates by a factor of 10 and set $\eta = 0.0025$, $\beta = 0.5$ and the initial distribution of weights to $J_{E,i} = J_I = 10$ for all $i$. For all eyelid response learning simulations, we chose $cf_{spont} = 1 Hz$ (Figs. 2, 4, 5).

**Error measure of learned Purkinje cell pause.** We defined the error between the PC pause and the $I_{target}$ (see Fig. 4, S3, S4 and S5) in the following way:

$$\epsilon_{tot} = \left(1 - \frac{\epsilon_{amp}}{h_{spont}}\right) + \frac{\epsilon_{fwhm}}{s} + 5 \cdot \frac{\epsilon_t}{s} \quad (22)$$

The first term depends on the amplitude of the PC pause relative to baseline firing, yielding a small error when the amplitude goes to zero. The second term corresponds to the normalized width of the PC pause. Finally, the third term is the normalized deviation of the pause's minimum from the target time, $\epsilon_t$. To increase the importance of this term, we scaled it by a factor 5. The error measure in Figs. S4 and S5 is the sum of $\epsilon_{tot}$ over all tested delays.

### Reduced CC model

The reduced synaptic model included only two synapse types. We also neglected facilitation and desensitization, yielding constant release probabilities and constant normalized quantal size:

$$u_{ij}^{slow}(t) = p_{v,slow}^{\alpha}$$

$$u_{ij}^{fast}(t) = p_{v,fast}^{\alpha}$$

$$q_{ij}(t) = 1. \quad (23)$$

**Table 3 | Synaptic parameters used in reduced model**

| | Drivers | Supporters |
|---|---|---|
| $N_{slow}$ | 3.5 | 4 |
| $N_{fast}$ | 14 | 6 |
| $p_{v,slow}$ | 0.8 | 0.4 |
| $p_{v,fast}$ | 0.6 | 0.2 |
| $\tau_{ref}^{slow}$ [ms] | 2000 | 2000 |
| $\tau_{ref}^{fast}$ [ms] | 20 | 20 |
| $p_{ref}$ | 0.6 | 0.6 |
| occurrence | 50% | 50% |

We obtain for the vesicle pool dynamics:

$$\dot{x}_{ij}^{slow}(t) = \frac{1 - x_{ij}^{slow}(t)}{\tau_{ref}^{slow}} - p_{v,slow}^{\alpha}(1 - p_{ref})x_{ij}^{slow}(t) \cdot m_j(t)$$

$$\dot{x}_{ij}^{fast}(t) = \frac{1 - x_{ij}^{fast}(t)}{\tau_{ref}^{fast}} - p_{v,fast}^{\alpha}x_{ij}^{fast}(t) \cdot m_j(t). \quad (24)$$

and the total synaptic weight becomes

$$W_{ij}(t) = N_{slow}^{\alpha} \cdot p_{v,slow}^{\alpha} \cdot x_{ij}^{slow}(t) + N_{fast}^{\alpha} \cdot p_{v,fast}^{\alpha} \cdot x_{ij}^{fast}(t). \quad (25)$$

Here the index $\alpha$ denotes membership in the driver or supporter category. The synaptic currents of the reduced model are computed as in the full model. Each GC receives exactly two driver and two supporter MF inputs with random and pairwise distinct identities. To eliminate any non-synaptic dynamics from the reduced model, we removed the GC membrane time constant yielding GC dynamics that follow the synaptic input instantaneously:

$$gc_i(t) = \alpha_i \cdot \max(I_{gc,i}(t) - \theta_i, 0). \quad (26)$$

Finally, GC threshold and gain adjustments were carried out similarly to the full CCM$_{STP}$ where instead of Eqs. (12) and (14) we used Eq. (23).

**Synaptic parameters of the reduced model.** The parameters of the reduced model were set to create two synapse types that capture the essence of the experimentally observed synaptic behavior: a strong and fast driver synapse, and a weak and slow supporter synapse. All synaptic parameters of the model used in Figs. 3, 4 and 6 are summarized in Table 3.

In Fig. 5, firing rates and release probabilities were randomly drawn from uniform distributions. In detail, the release probabilities of the slow pool, $p_{v,slow}$, were drawn from distributions with a lower and upper bound of 0.1 and 0.9, respectively, (Fig. 5a, d, g, and h), and the corresponding release probabilities of the fast pool were calculated according to $p_{v,fast} = \frac{2}{3}p_{v,slow}$, keeping them strictly lower. The lower and upper bounds of the distribution of firing rates used in panels **a** and **d** were 5 Hz and 270 Hz, resulting in firing rate standard deviations of $\sigma_{rate} \approx 38.2$ Hz for the two-groups case (Fig. 5a) and $\sigma_{rate} \approx 15.3$ Hz for the five groups case (Fig. 5d). The bounds of the distributions in panels **g** and **h** were chosen to match the average group firing rates equal to those in panel **d** and firing rate standard deviations that increased with the group index, i.e. $\sigma_{rate} \approx \{5.0, 7.6, 10.2, 12.7, 15.3\}$ Hz for groups 1 to 5, respectively. Finally, the sizes of the slow vesicle pool were fixed at $N_{slow} = 4$ and the size of the fast vesicle pools were set to decrease with the group index, i.e. $N_{fast} = \{16, 6\}$ for the two-groups case, and $N_{fast} = \{16, 12, 8, 6, 6\}$ for the five groups case. Finally, the desired rank correlation between $p_v$ identities and MF identities was achieved by creating a Gaussian copula reflecting their statistical dependency and reordering the marginal $p_v$ and MF distributions accordingly.

**Derivation of $\tau_{syn}$ and $A_t$**

In the reduced model, we derived an analytical solution to the synaptic current driving a GC in response to the CS. Since the equations describing slow and fast vesicle pool dynamics are formally very similar, we describe the derivation for a single slow pool only. Additionally, we suppress all indices for the sake of readability. We assume that the MF rate $m(t)$ switches instantaneously from $m_{preCS}$ to $m_{CS}$ at time $t' = 0$. Integration of equations (Eq. (24)) from $t' = 0$ to $t$ yields:

$$x(t) = (x_{preCS}^* - x_{CS}^*) \exp\left(-\left(\frac{1}{\tau_{ref}} + p_v(1 - p_{ref})m_{CS}\right)t\right) + x_{CS}^*, \quad (27)$$

Here, $x_{preCS}^*$ and $x_{CS}^*$ denote the steady-state values of x before (preCS) and after (CS) the firing rate switch. They are given by

$$x_\gamma^* = \frac{1}{1 + \alpha p_v m_\gamma}, \quad (28)$$

with

$$\alpha = \begin{cases} \tau_{ref}(1 - p_{ref}) & \text{for slow pool} \\ \tau_{ref} & \text{for fast pool} \end{cases} \quad (29)$$

Equation (27) defines the synaptic time constant that governs the speed of transition from a steady-state value before the CS to a steady-state value during the CS:

$$\tau_{syn} = \tau_{ref} \cdot x_{CS}^* = \frac{\tau_{ref}}{1 + \alpha p_v m_{CS}} \quad (30)$$

This equation is similar to one derived previously[55,87]. The total synaptic current per unit time for a single pool during the CS is given by

$$I_{syn}(t) = N p_v x(t) m_{CS} \quad (31)$$

Combining Eqs. (27) and (31) we obtain

$$I(t) = \frac{N p_v m_{CS}}{1 + \alpha p_v m_{CS}}\left[1 + \frac{\alpha p_v(m_{CS} - m_{preCS})}{1 + \alpha p_v m_{preCS}}\exp\left(-\frac{t}{\tau_{syn}}\right)\right]$$

$$= \underbrace{A_s}_{\text{steady state}} + \underbrace{A_t}_{\text{transient amplitude}} \exp\left(-\frac{t}{\tau_{syn}}\right) \quad (32)$$

Thus, the transient amplitude for a single vesicle pool is

$$A_t = \frac{N p_v m_{CS}}{1 + \alpha p_v m_{CS}} \frac{\alpha p_v(m_{CS} - m_{preCS})}{1 + \alpha p_v m_{preCS}} \quad (33)$$

For a single synapse, the total transient amplitude is the sum of the individual fast pool and slow pool transients:

$$A_t^{tot} = A_t^{slow} + A_t^{fast} \quad (34)$$

To generate the surface plots in Fig. 4 and Fig S3 we generated $10^5$ firing rates from the driver and supporter MF rate distributions, respectively, and used Equations (30), (33) and (34) to calculate the corresponding values of the $A_t$ and $\tau_{syn}$. From these, the plots of the joint $A_t$ and $\tau_{syn}$ distribution and the marginal distributions were generated using a two- or one-dimensional kernel density estimator, respectively[88]. Note that, formally, $\tau_{syn}$ is maximal when $m_{CS} = 0$. In that case, however, there is no synaptic transmission as $A_t^{tot} = A_t^{slow} = A_t^{fast} = 0$. When plotting the joint $A_t$-$\tau_{syn}$ distribution in Fig. 4 and Fig S3, we therefore omitted time constants and transient amplitudes corresponding to $m_{CS} = 0$.

## Bayesian estimation of time intervals

To learn the mapping between $t_m$ and $t_e$, we presented CCM$_{STP}$ with variable intervals drawn from various prior distributions ($t_s$) subjected to measurement noise. The interval was introduced as a tonic input to MFs, similar to the CS in the eyelid simulations. The onset of this tonic input caused an abrupt switch of the MF input rates that persisted over the course of a trial. For each iteration of our learning algorithm, we generated target signals sampled randomly from one of five different uniform prior distributions: 25–150 ms, 50–200 ms, 100–300 ms, 200–400 ms, 300–500 ms. Learning was carried out separately for each interval and for 12000 iterations. We found that to achieve the correct biases for the two longest intervals, we had to introduce a higher CF baseline firing rate, $cf_{spont} = 5$ Hz. The other learning parameters were kept the same as in the eyelid learning simulations.

In keeping with ref. 33, we modeled the DN neuron as an integrator, whose rate was calculated according to

$$dn(t) = \int \left( I_{ext} - J_{pc} pc(t) \right) dt, \tag{35}$$

where the $J_{pc}$ is the weight of the inhibitory PC-DN synapse and $I_{ext} = \langle pc \rangle$ is an external excitatory input to DN. It was set equal to the average PC firing rate during the interval period to ensure that excitation and inhibition onto the DN are of comparable size. For simplicity, we set $J_{pc} = 1$.

In order to map the DN rate to a time axis (Fig. 6f, j), we rescaled every individual DN output curve according to:

$$\widehat{dn}(t) = \left( t_{s,max} - t_{s,min} \right) \frac{dn(t) - dn_{min}}{dn_{max} - dn_{min}} + t_{s,min}, \tag{36}$$

where $t_{s,max}$ and $t_{s,min}$ are the maximum and minimum of the respective prior interval and $dn_{max}$ and $dn_{min}$ are the maximum and minimum values of the DN firing rate. Since the transformation described in Eq. (36) is linear, the essential features exhibited by the DN firing rate (i.e. its biases) are preserved.

To show how the theoretical Bayesian least squares (BLS) interval estimate can be obtained, we follow the reasoning from ref. 33. It is assumed that to estimate a time-interval, $t_s$, subjects perform a noisy measurement, $t_m$, according to:

$$p(t_m|t_s) = \frac{1}{\sqrt{2\pi(w_{weber}t_s)^2}} e^{-\frac{(t_s - t_m)^2}{2(w_{weber}t_s)^2}}. \tag{37}$$

Note that the standard deviation of the estimate of $t_m$ increases with the length of the interval $t_s$ with proportionality factor $w_{weber}$, which is the weber fraction. Given the prior distribution of time intervals, $\Pi(t_s)$, the Bayesian estimate of $t_s$ given $t_m$ is:

$$p(t_s|t_m) \propto \Pi(t_s)p(t_m|t_s). \tag{38}$$

The BLS estimate is the expected value of the previous expression:

$$t_e = E[p(t_s|t_m)]. \tag{39}$$

We performed a least squares fit of the BLS model to the CCM$_{STP}$ outputs (from all five interval distributions simultaneously) with $w_{weber}$ as a single free parameter.

## Recurrent Golgi cell inhibition

To probe the effect of recurrent inhibition in the reduced CCM$_{STP}$, we added one Golgi cell (GoC) that received excitatory inputs from all GCs and formed inhibitory synapses onto all GCs. For simplicity, we

assumed that the GoC fires with a rate $goc$ equal to the average GC firing rate, similarly to the MLI, and that all GoC to GC synapses have identical weights, $J_{goc}$:

$$goc(t) = \frac{1}{N} \sum_{i=1}^{N} gc_i(t) = \langle gc(t) \rangle$$
$$I_{gc,i}(t) = \sum_{j \in K} W_{ij}(t)m_j(t) - J_{goc} \cdot goc(t) \tag{40}$$
$$gc_i(t) = \alpha_i \cdot \max(I_{gc,i}(t) - \theta_i, 0).$$

The above equations imply that, in this configuration, the GoC acts as an activity-dependent GC threshold.

To ensure that the overall GC activity level in the reduced CCM$_{STP}$ with GoC inhibition is comparable to the case without, we require the same criterion as above: an average GC rate of 5 Hz and a fraction of activated GCs of 0.2 in steady state. Since the average GC input now depends on the average GC firing rate itself, manual adjustment of GC thresholds, $\theta_i$, and gains, $\alpha_i$, carried out as above, is not feasible.

Instead, a steady-state solution of the set of Eq. (40) satisfying our requirements has to be found numerically. We first set up the CC network without the GoC and adjusted GC thresholds, $\theta_i$, and gains, $\alpha_i$, according to the procedure described above. Note that in the reduced model, due to every GC receiving the same combination of inputs (i.e. 2 supporter and two driver inputs), both $\theta_i$ and $\alpha_i$ are similar across GCs. We thus made the additional simplification of setting $\theta = E(\theta_i)$ and $\alpha = E(\alpha_i)$ for all GCs. We then reduced GC thresholds by 10% and introduced the GoC.

To obtain the average steady-state GC firing rate we assumed that the synaptic currents of a single GC are normally distributed across MF input patterns or, equivalently, across GCs. Mean and variance of the GC inputs are:

$$\left\langle I_{gc}^* \right\rangle = E\left( I_{gc,i}^* \right) = E\left( \sum_{j \in K} W_{ij}^* \cdot m_j \right) - J_{goc} \cdot \langle gc^* \rangle$$
$$\sigma_I^2 = \mathrm{Var}\left( I_{gc,i}^* \right) = \mathrm{Var}\left( \sum_{j \in K} W_{ij}^* \cdot m_j \right) \tag{41}$$

We can then express the average GC firing rate in the $N \to \infty$ limit as:

$$\langle gc^* \rangle = \alpha \int_{-\infty}^{+\infty} \max\left( \left\langle I_{gc}^* \right\rangle + \sigma_I^* \cdot \xi - \tilde{\theta}, 0 \right) \exp\left( -\frac{\xi^2}{2} \right) \frac{d\xi}{\sqrt{2\pi}} \tag{42}$$

where $\widetilde{\theta} = 0.9\theta$. The fraction of active GCs $f$ can be written as:

$$f = \frac{1}{2} \mathrm{erfc}\left( \frac{\theta - \left\langle I_{gc}^* \right\rangle}{\sqrt{2}\sigma_{I^*}} \right) \tag{43}$$

We can now impose that

$$\langle gc^* \rangle = 5\mathrm{Hz}$$
$$f = 0.2 \tag{44}$$

and find a self-consistent solution of Eqs. (41), (42), and (43) by adjusting the parameters $J_{goc}$ and $\alpha$. To do so we used the hybrid numerical root-finder from the GNU scientific library[89] with default step size.

## Reporting summary

Further information on research design is available in the Nature Portfolio Reporting Summary linked to this article.

## Data availability
No experimental data were generated in this study.

## Code availability
Figures were generated with Matlab (R2019b) and python (3.8). All simulations were performed with C++11 using the GNU scientific library (2.6)[89] and the armadillo library (11.0.1)[90]. The code is available on the following GitHub repository: https://github.com/alessandrobarri/cerebellar_cortex_input_STP.

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

## Acknowledgements

A.B. thanks Gianluigi Mongillo and Zuzanna Piwkowska Zvonkine for helpful discussions. We thank the DiGregorio Lab for feedback on this manuscript. This work is supported by the Institut Pasteur, Centre National de la Recherche Scientifique, Fondation pour la Recherche Médicale (FRM EQU202003010555), Fondation pour l'Audition (FPA-RD-2018-8), BioPsy Laboratory of Excellence, and the Agence Nationale de la Recherche (ANR-17-CE16-0019, and ANR-18-CE16-0018, ANR-19-CE16 0019-02, ANR-21-CE16-0036-01), which were awarded to the laboratory of DAD.

## Author contributions

All simulations and analyses were performed by A.B. A.B., M.W., M.J., and D.A.D. conceived the project and wrote the manuscript.

## Competing interests

The authors declare no competing interests.
