## [Peer Review File · Nature Communications]

Synaptic basis of a sub-second representation of time in a neural circuit modelREVIEWER COMMENTS

Reviewer #1 (Remarks to the Author):

The basic idea of this manuscript is that a heterogeneity of synaptic time constants---arising from a variety of biophysical mechanisms---providing input to cerebellar granule cells are sufficient to enable granule cells to exhibit temporal basis functions. The computational model is quantitatively constrained by a large body of empirical data. To my knowledge, this is the first paper drawing out the possibility that short-term plasticity could be important in constructing a well-behaved temporal basis set in cerebellum (or anywhere else for that matter). There is quite broad interest in the neural basis of timing information including in studies of episodic memory, working memory, and perception. Because this paper expands the hypothesis space for construction of temporal basis sets with an elegant and biophysically realistic model, it makes a strong contribution and is appropriate in my view for Nature Communications.

Specific reactions:

It's nice that the model can be connected to behavior. However, I am unable to evaluate whether the gradient descent model for Purkinje cell firing (and classical conditioning performance) is widely-used or whether it's biologically reasonable. Other reviewers may have additional insights into this question. The illustration of Weber-like properties of this model largely follow from a well-chosen temporal basis set; as long as the learning rule at granule cell to Purkinje cells doesn't introduce a strong scale, this should result. It may be worth pursuing the question (perhaps in future work) how sensitive this is to the distribution of time constants. My guess would be that Weber law becomes precise to the extent the distribution goes like τ^{-1} .

Specific expository points:

- * Fewer acronyms would make the paper easier to read.
- * Lots of people are interested in time and timing. Audience would be expanded with a gentle tutorial to the "standard model" of cerebellar organization.
- * Even if one sticks with lots of acronyms, STP should be spelled out on first usage.

Reviewer #2 (Remarks to the Author):

Barri and colleagues develop a biologically-based model of how the cerebellum may represent time. Previous work by this lab has characterized and quantified short-term synaptic plasticity (STP) at the MF-Gr synapse. Here a feed-forward model of the cerebellar circuitry MF-Gr-PC is developed in which biologically derived STP at the MF-Gr synapses imposes exponentially decaying (for the most part) activity patterns in response to tonic steps of MF input, and these exponentially decaying basis functions are used by the PC neurons create late responses (pauses) by adjusting the Gr-PC weights. The model shows that this mechanism can account well for PC pauses with delays up to 700 ms. The paper carefully quantitatively characterizes the mechanisms, parameter dependencies of the model, and examines if the temporal representation at PCs is Bayesian. Although there are issues that need to be addressed, the development of a STP based model of cerebellar timing that is strongly grounded in experimental data is a valuable contribution to our understanding of cerebellar function and timing.

Figure 6. The presentation of the Ready-Set-Go (RSG) task is rather confusing, and a bit misleading. The main reason being that the model is not actually solving the RSG task, but encoding the RS interval in a Bayesian fashion. Indeed, an STP based model seems to be not well suited to solve both the sensory and motor components of the RSG task. Since the RSG task is not actually being solved, Fig 6 should just focus on the encoding of the RS interval, basically the same task as in earlier figures, and in the text the authors can interpret this in the context of the RSG task. This section should be explained better.

If they focus on the RSG task, the authors should explicitly state whether they think the CCMstp model does/does not account for the entire RSG task.

A strength of the paper is the degree to which it is grounded in the experimental data. Thus, the use of a gradient descent based learning rule at the PC cell detracts from the strength of the paper. If a nonbiological learning rule is required in this model, but not in network based models (Mauk), this would seem to pose an argument against the current model.

Mathematically, the model, and the reduced model in particular, seems to be mathematically equivalent to the decaying exponential/Laplacian models of Howard (1,2). Some of this work is cited. The current work is embedded in a much more solid biological framework, but given that the models are perhaps a mathematically equivalent, this should be discussed in more detail.

There is a section in the Discussion on predictions of the model, but actual testable predictions are not carefully spelled out. One clear prediction would seem to be that when recording from Gr cells the responses should be primarily exponentially decaying. Based on Kennedy et al (3), one might argue that this prediction is not met. But I suppose one could argue that given that late spiking Gr cells are the

minority that the model is consistent with the data. A more serious issue may be that a significant number of Gr cells seem to have multiple peaks—an observation that would seem to better accounted for by network based models of timing (Mauk).

The network based Gr-Go-Gr model of timing requires multiple Go units, thus one cannot make any strong statements based on a model with one Go unit. The diversity of Gr unit profiles requires a diversity of Go unit profiles. Thus this model (Fig S6) should be expanded to include multiple Go units or the statement regarding this model (along with the figure) removed from the Discussion.

A question that arises from the current model is that if the purpose of STP at the MF-Gr synapse is to create Gr temporal basis functions for timing, why would one not observe more short-term facilitation at this synapse?

Figure 3d. I think the p values in the panel should read p_v.

Fig 4. Why is the error (last column) of the first row labeled as “error” and in the other rows “rel error”? It would be helpful if the axes were the same across all rows.

In Fig 5a,d,g the lower panel should be labeled p_v.

1 Tiganj, Z., Hasselmo, M. E. & Howard, M. W. A Simple biophysically plausible model for long time constants in single neurons. *Hippocampus* 25, 27-37, (2015).

2 Liu, Y., Tiganj, Z., Hasselmo, M. E. & Howard, M. W. A neural microcircuit model for a scalable scale-invariant representation of time. *Hippocampus* 29, 260-274, (2019).

3 Kennedy, A. et al. A temporal basis for predicting the sensory consequences of motor commands in an electric fish. *Nat Neurosci* 17, 416-422, (2014).

Reviewer #1 (Remarks to the Author):

The basic idea of this manuscript is that a heterogeneity of synaptic time constants---arising from a variety of biophysical mechanisms---providing input to cerebellar granule cells are sufficient to enable granule cells to exhibit temporal basis functions. The computational model is quantitatively constrained by a large body of empirical data. To my knowledge, this is the first paper drawing out the possibility that short-term plasticity could be important in constructing a well-behaved temporal basis set in cerebellum (or anywhere else for that matter). There is quite broad interest in the neural basis of timing information including in studies of episodic memory, working memory, and perception. Because this paper expands the hypothesis space for construction of temporal basis sets with an elegant and biophysically realistic model, it makes a strong contribution and is appropriate in my view for Nature Communications.

We thank the reviewer for the encouraging words.

Specific reactions:

It's nice that the model can be connected to behavior. However, I am unable to evaluate whether the gradient descent model for Purkinje cell firing (and classical conditioning performance) is widely-used or whether it's biologically reasonable. Other reviewers may have additional insights into this question. The illustration of Weber-like properties of this model largely follow from a well-chosen temporal basis set; as long as the learning rule at granule cell to Purkinje cells doesn't introduce a strong scale, this should result. It may be worth pursuing the question (perhaps in future work) how sensitive this is to the distribution of time constants. My guess would be that Weber law becomes precise to the extent the distribution goes like τ^{-1} .

We address concerns about learning rule in response to comments of reviewer 2.

Specific expository points:

* Fewer acronyms would make the paper easier to read.

We understand the frustration of the reviewer as the number of acronyms can be challenging. However, nearly all the acronyms are commonly used in the literature. We did however eliminate "CC" and defined the first usage of STP (that was mistakenly left out in the first version of the manuscript). We kept CCM_{STP} as it is the name of our model. We also removed the acronym for molecular layer interneurons in the main text but defined an acronym (MLI) in the methods section, where it was referred to several times.

* Lots of people are interested in time and timing. Audience would be expanded with a gentle tutorial to the "standard model" of cerebellar organization.

We extended the introduction with a few sentences that describe the anatomical organization of the cerebellar cortex. The relevant paragraph now reads (pg. 3 parag. 3):

"The cerebellar cortex is a prototypical microcircuit known to be important for generating temporally precise motor[1] and cognitive behaviors[2-5] on the sub-second timescale. It receives mossy fibers (MFs) from various sensory, motor and cortical areas. MFs are thought to convey contextual information and converge onto granule cells (GCs), the most numerous neuron in the brain. The excitatory GCs project onto the inhibitory molecular layer interneurons and Purkinje cells (PCs). PCs, being the sole output neurons of the cerebellar cortex, inhibit neurons in the deep cerebellar

nuclei. According to the Marr-Albus-Ito model of cerebellar cortical circuit computations, precisely timed Purkinje cell activity can be learned by adjusting the synaptic weights between GCs with differing activity patterns[6,7]. This largely feed-forward circuitry has been proposed to learn the temporal contingencies required for prediction from neural sequences across the population of GCs within the input layer[8]. The synapses between MFs and GCs are highly variable in their synaptic strength and STP time course[9]. Therefore, we hypothesized that STP of MF-GC synapses could be used as internal timers for a population clock within the cerebellar cortex to generate neural dynamics necessary for temporal learning.”

* Even if one sticks with lots of acronyms, STP should be spelled out on first usage.

This is a good point; we now define the acronym “STP” the first time it appears in the introduction.

Reviewer #2 (Remarks to the Author):

Barri and colleagues develop a biologically-based model of how the cerebellum may represent time. Previous work by this lab has characterized and quantified short-term synaptic plasticity (STP) at the MF-Gr synapse. Here a feed-forward model of the cerebellar circuitry MF-Gr-PC is developed in which biologically derived STP at the MF-Gr synapses imposes exponentially decaying (for the most part) activity patterns in response to tonic steps of MF input, and these exponentially decaying basis functions are used by the PC neurons create late responses (pauses) by adjusting the Gr-PC weights. The model shows that this mechanism can account well for PC pauses with delays up to 700 ms. The paper carefully quantitatively characterizes the mechanisms, parameter dependencies of the model, and examines if the temporal representation at PCs is Bayesian. Although there are issues that need to be addressed, the development of a STP based model of cerebellar timing that is strongly grounded in experimental data is a valuable contribution to our understanding of cerebellar function and timing.

We thank the reviewer for the encouraging words.

Figure 6. The presentation of the Ready-Set-Go (RSG) task is rather confusing, and a bit misleading. The main reason being that the model is not actually solving the RSG task, but encoding the RS interval in a Bayesian fashion. Indeed, an STP based model seems to be not well suited to solve both the sensory and motor components of the RSG task. Since the RSG task is not actually being solved, Fig 6 should just focus on the encoding of the RS interval, basically the same task as in earlier figures, and in the text the authors can interpret this in the context of the RSG task. This section should be explained better.

We fully agree with the reviewer and have modified the figure and the associated text to address the concern. The RSG task has been used widely because the SG (produced interval) serves as a direct behavioral readout for how the RS (sample interval) was encoded. However, as the reviewer has noted, the focus of our paper is not on RSG but rather on the Bayesian encoding of the sample interval (RS). Specifically, we now use RSG only to make the point that the encoding of time intervals in the presence of a prior distribution is subject to Bayesian integration. We then focus the question on how such Bayesian integration may come about (independent of the RSG) and explain how our model provides a natural solution for this computation. As part of this revision, we have also modified the relevant panel in Figure 6 to highlight the Bayesian interval estimation – not the ready-set-go task.

The text describing the Bayesian interval estimation is (pg. 17, parag. 2):

“A recent study developed a cerebellar model called TRACE for temporal Bayesian computations³³. TRACE implements Bayesian integration by incorporating two features. First, it assumes that GCs form a temporal basis set that exhibits temporal scaling. This feature accounts for the scalar variability of timing. Second, it assumes that prior-dependent learning alters the GC-PC synapses. This feature allows the dentate nucleus neurons (DNs) downstream of PCs to represent a Bayesian estimate of the time interval.

In our analysis of eyelid conditioning (**Fig. 2**), we showed that CCM_{STP} generates PC firing rate pauses whose width and amplitude scale with time (**Fig. 6a**). Therefore, we reasoned that CCM_{STP} might have the requisite features for Bayesian integration. To test this possibility quantitatively, we presented our model with variable intervals drawn from various prior distributions. The interval was introduced as a tonic input to MFs, similar to the CS in the eyelid simulations. The onset of this tonic input caused an abrupt switch of the MF input rates that persisted over the course of a trial. During learning, we subjected the model to intervals sampled randomly from a desired prior distribution.

We tested CCM_{STP} with five different uniform distributions of ready-set intervals (25-150 ms, 50-200 ms, 100-300 ms, 200-400 ms, 300-500 ms), resulting in PC pauses that broadened for longer interval distributions, and integrated DN activity that could easily match the Bayesian least-square model³³ by adjusting a single parameter, the Weber fraction w_{weber} (see Methods; **Fig. 6d,h**). The reduced model interval estimates were more similar to the Bayesian estimates than for CCM_{STP} with native synaptic parameters, especially for the 200-400 ms and 300-500 ms intervals (**Fig. 6h-k**). Nevertheless, the CCM_{STP} simulations show that a GC basis generated by MF-GC STP is sufficient for driving Bayesian-like learning of time intervals spanning several hundreds of milliseconds. It should be noted that our GC temporal basis was not explicitly constructed to accommodate scalar properties. Nevertheless, as in the TRACE model, we observed that interval estimates were biased towards the mean and that these biases were larger for longer intervals. These results suggest that a GC basis set generated from the diverse properties of native MF-GC synapses likely exhibits a scalar property necessary for generating optimally timed behaviors.”

If they focus on the RSG task, the authors should explicitly state whether they think the CCMstp model does/does not account for the entire RSG task.

As the reviewer has astutely noted, our work only bears on the computational mechanisms underlying Bayesian integration and not the ensuing production of that interval. Indeed, the later could involve integration of signals in the cerebellum with other effector-specific motor regions on the brain that go beyond the scope of our model.

A strength of the paper is the degree to which it is grounded in the experimental data. Thus, the use of a gradient descent based learning rule at the PC cell detracts from the strength of the paper. If a nonbiological learning rule is required in this model, but not in network based models (Mauk), this would seem to pose an argument against the current model.

We agree with the reviewer that the nature of the cerebellar learning rule employed is of crucial importance. We would like to point out that our gradient descent (GD) learning rule is a variant of that used in the modelling study by Clopath et al. 2014¹, which, at least for VOR learning, is consistent with the literature²⁻⁴. As explained in the Methods section, we explicitly compute a CF rate that is used to update the GC-PC weights. In contrast to Clopath et al. 2014¹, we constrained the CF rate to be strictly positive, since the CF rate cannot be negative.

The deterministic learning rule we employed provides directional information about the GC-PC weight adjustment through and a gradual reduction of the CF spiking rate that is concomitant with the progression of learning. This choice of learning rule is supported by the experimental evidence for graded CF spikes^{5,6}, for directional CF information⁷ and for the disappearance of US triggered CF spikes⁸. All of these phenomena are consistent with a GD rule. Furthermore, a recent study that thoroughly investigated the role of the CF in cerebellar learning found that the GC- and CF-spike pairings necessary for the induction of LTD/LTP under physiological conditions are compatible with a stochastic GD rule⁹. Thus, our rule can be seen as a deterministic variant of the stochastic GD rule proposed in that paper.

We are, however, aware that there are open questions as to whether cerebellar learning conforms to a GD rule, and if so, how GD is implemented mechanistically.

We added the following section entitled “Choice of the cerebellar learning rule” to the Discussion (pg. 22):

“The learning rule we used here was adapted from a previous modeling study that investigated cerebellar adaptation of the vestibular ocular reflex and was argued to be biologically plausible[72].

This synaptic weight update rule is mathematically equivalent to a gradient descent in which the error magnitude is transmitted via the climbing fiber[72]. Consequently, our rule features graded climbing-fiber responses and a gradual reduction in climbing-fiber spiking that is concomitant with the progression of learning. These phenomena have been experimentally observed[43,73]. Moreover, a recent study that thoroughly investigated the role of the climbing fiber spike in cerebellar learning found that the GC and climbing-fiber spike pairings necessary for the induction of long-term depression /potentiation under physiological conditions are compatible with a stochastic gradient descent rule[46]. Our rule can be seen as a deterministic variant of this.”

Mathematically, the model, and the reduced model in particular, seems to be mathematically equivalent to the decaying exponential/Laplacian models of Howard (1,2). Some of this work is cited. The current work is embedded in a much more solid biological framework, but given that the models are perhaps a mathematically equivalent, this should be discussed in more detail.

We added a paragraph in the Discussion that better explains the most important differences between Marc Howard’s work and ours. The passage reads:

“Recent theoretical work proposes that a scale-invariant neuronal representation of a temporal stimulus sequence can be obtained by using a population of leaky integrators that produce exponentially decaying neural activity transients[52]. Indeed, exponential-like activity has been observed in the entorhinal cortex – a region a region that projects to hippocampus. The exponential-like population activity is reminiscent of the GC temporal basis set in CCM_{STP} following a persistent firing rate changes. However, the MF-GC synaptic inputs are always a mixture of multiple exponential components. Nevertheless, our work suggests that STP could be a plausible biological mechanism explaining the occurrence of exponential dynamics in neuronal populations[6] and merits further theoretical and experimental investigation.”

There is a section in the Discussion on predictions of the model, but actual testable predictions are not carefully spelled out. One clear prediction would seem to be that when recording from Gr cells the responses should be primarily exponentially decaying. Based on Kennedy et al (3), one might argue that this prediction is not met. But I suppose one could argue that given that late spiking Gr cells are the minority that the model is consistent with the data. A more serious issue may be that a significant number of Gr cells seem to have multiple peaks—an observation that would seem to better accounted for by network based models of timing (Mauk).

To the best of our knowledge, the Kennedy paper shows MF and GC activity during a typical EOD. The sensory activation resulting from the EOD generation are dynamic throughout a 200 ms time window. In contrast, our model describes exponential-like GC responses (Figs. 2-4), but in response to a persistent switch in MF activity. We expect GC activity generated by MF-GC STP to be more complex in response to time-varying MF inputs similar to those observed in Kennedy et al. Therefore, it is hard to directly compare the GC activity of our model to the recordings from Kennedy et al. Moreover, the diversity of GC activity in the Kennedy study seems to be inherited from the substantial temporal heterogeneity in MF dynamics (see Fig. 1e in Kennedy et al.).

Furthermore, we agree with the reviewer that GC activity featuring delays, multiple peaks and pauses is not easily accounted for by STP alone, but could more easily be explained by mechanisms involving UBCs and/or Golgi-cells. However, slower synaptic facilitation time constants could produce such activity patterns as well. Nevertheless, we expect STP diversity to contribute significantly to the generation of a rich temporal basis of GC activity, but we also expect complementary contributions from unipolar brush cells (only in certain areas of cerebellum) and Golgi-cell feedback. In support of

the mechanistic complementarity, we showed in Supplementary Figure 6 that the addition of (single) Golgi-cell inhibition increases the variability in the initiation and time of peak of GC activity (with some peaking as late as 200 ms). (Regarding the difference between or Golgi cell implementation and Mauk-type models, see the responses to other comments below.)

To more explicitly describe our predictions for the GC activity, **we added a new paragraph to the Discussion section**. Specifically, we remind the reader that our model predicts that step-like change in MF activity (using optogenetic or electrical stimulation) should generate a distribution of GC firing rate decays. Experimental evidence that GC activity profiles are exclusively exponentially decaying would be consistent with the simplest version of our model (i.e. depression dominated STP), while the appearance of GCs with multiple peaks would suggest the presence of more prominent facilitation or a more network based mechanism (i.e. Golgi). Our expectation is that GC activity is probably a mixture of both. Despite this prediction, measurement of population activity at high temporal resolution is currently not possible.

The modified paragraph in the Discussion now reads (pg. 21, parag. 3):

“CCM_{STP} predicts that persistent changes in MF activity would generate exponential-like GC activity profiles (**Figs. 2, 4**). However, although the majority of simulated GCs shown here are active at the onset of the CS, this is not a necessary feature of CCM_{STP}. When we included a single, average-subtracting Golgi cell (possibly representing the “common mode” of Golgi Cell population activity [63]), more GCs showed a delayed onset firing and the variability of onset and peak times (**Fig. S6**). Note that our implementation of Golgi cell feedback is simplified and does not account for reciprocal inhibition between multiple Golgi cells, which in simulations has also been shown to generate diverse GC activity [12, 29]. To test these predictions, MFs could be driven at constant rates using direct electrical or optogenetic stimulation of the cerebellar peduncle *in vivo* or the white matter in acute brain slices, with an without intact Golgi cell inhibition. Unfortunately, high-temporal resolution population recordings of GCs are challenging due to the small size of GC somata. Perhaps in the future small impedance silicon probe recordings[69] or ultra-fast optical indicators[70] might permit testing our hypotheses experimentally. If successful, we predict that the time course of GC responses should be diverse and exponential-like, with prominent delayed activity in some granule cells when Golgi cells are intact. Furthermore, decreasing or increasing the MF firing rate should in turn slow or accelerate GC responses, respectively. It should be noted that the addition of a Golgi cell in the model did not change the synaptic time constant distribution and therefore did not improve simulated delayed eyelid conditioning. Finally, for complex behavioral experiments in which the MF activity is dynamic (and measurable), one could examine which circuit connectivity of the CCM_{STP} best reproduces the measured GC activity.”

The network based Gr-Go-Gr model of timing requires multiple Go units, thus one cannot make any strong statements based on a model with one Go unit. The diversity of Gr unit profiles requires a diversity of Go unit profiles. Thus this model (Fig S6) should be expanded to include multiple Go units or the statement regarding this model (along with the figure) removed from the Discussion.

The reviewer is correct that we cannot make any statements about network-based mechanisms of the Mauk type with only a single Golgi-cell. We have removed the misleading text:

“Computer simulations have shown that recurrent GC-GoC-cell network dynamics could also generate a rich GC temporal basis set[8,21]. However, experimental support of such circuit dynamics is lacking.

When we included recurrent GC-GoC synapses in our simulations, more GCs showed a delayed onset firing (**Fig. S6**), but the synaptic time constant distribution and learning rate remained unaltered. These simulations are consistent with a “common mode” GoC cell population activity[20].

Our goal was to show that a majority of simulated GCs being active at the onset of the CS (Figs. 2, 4) is not a necessary feature of our model, but that delayed onset of GC firing can be obtained by including simplified Golgi cell feedback. We believe that investigating the role of MF-GC STP in the context of Mauk-type recurrent GC-Golgi-Cell-cell network models is beyond the scope of this study, but that it is an interesting subject for future research. We stated this in the discussion.

The relevant section now reads (Pg. 21, parag. 1):

“Consistent with the importance of MF-GC STP, delayed eyelid conditioning was selectively altered as a result of the loss of fast EPSCs in AMPAR KO mice[65]. Simulations including realistic NMDA, and spillover dynamics[68] can further enrich the temporal scales available to the network[67]. **It would be of particular interest to investigate the role of MF-GC STP in the context of recurrent GC-Golgi-Cell-cell network models that have been shown to generate rich GC temporal basis sets[12,29].** Finally, we note that MF-GC STP and other timing mechanisms described above are not mutually exclusive, but presumably act in concert with the diverse intrinsic properties of GCs[68] and PCs[69] in order to cover different timescales or increase mechanistic redundancy.”

We now mention our Golgi Cell simulations in the “Predictions of the CCM_{STP}” section where we clarify the intent and scope of these (see above).

A question that arises from the current model is that if the purpose of STP at the MF-Gr synapse is to create Gr temporal basis functions for timing, why would one not observe more short-term facilitation at this synapse?

We do not know why the fraction of facilitating synapses is only approximately 20% in the vestibular cerebellum¹¹. We have constrained our model with the data from ref.¹¹ and our conclusions are thus consistent with the STP diversity that was experimentally observed in lobule X. It is possible that other areas of the cerebellum feature more prominent short-term facilitation. However, at the current stage, we could only speculate about the computational role of synaptic depression- versus facilitation-dominated STP in cerebellar temporal learning (e.g. further increase the diversity of GC activity patterns). We are not entirely convinced that this discussion will add value to the article. However, if the reviewer insists that given some perspective about the possible role of short-term facilitation is necessary we are happy to consider it in the next round of revision.

Figure 3d. I think the p values in the panel should read p_v.

We corrected the legend of figure 3d accordingly.

Fig 4. Why is the error (last column) of the first row labeled as “error” and in the other rows “rel error”? It would be helpful if the axes were the same across all rows.

Our choice of the y-axis label (“rel. error”) is indeed confusing. The aim of this figure is the following: The last panel of the first row shows the error measure associated with the PC eye-lid responses for the parameter set that we consider the control case. The last panel in all subsequent rows shows how this error changes for the other cases. Specifically, a value of 1 means no change, a value > 1 means worse performance and a value < 1 means better performance relative to the control case. We believe that this way to present the results is more informative.

Accordingly, we changed the y-axis label to “change of error (%)”. If the reviewer finds this confusing still, we can of course show the absolute error in all panels, as suggested.

In Fig 5a,d,g the lower panel should be labeled p_v .

We added the symbol p_v to the relevant panels in figure 5.

1. Clopath, C., Badura, A., De Zeeuw, C. I. & Brunel, N. A Cerebellar Learning Model of Vestibulo-Ocular Reflex Adaptation in Wild-Type and Mutant Mice. *Journal of Neuroscience* **34**, 7203–7215 (2014).
2. Ito, M. Long-Term Depression. *Annual Review of Neuroscience* **12:85-102**, 18 (1989).
3. Jörntell, H. & Hansel, C. Synaptic Memories Upside Down: Bidirectional Plasticity at Cerebellar Parallel Fiber-Purkinje Cell Synapses. *Neuron* **52**, 227–238 (2006).
4. Gao, Z., van Beugen, B. J. & De Zeeuw, C. I. Distributed synergistic plasticity and cerebellar learning. *Nature Reviews Neuroscience* **13**, 619–635 (2012).
5. Mathy, A. *et al.* Encoding of Oscillations by Axonal Bursts in Inferior Olive Neurons. *Neuron* **62**, 388–399 (2009).
6. Najafi, F. & Medina, J. F. Beyond “all-or-nothing” climbing fibers: graded representation of teaching signals in Purkinje cells. *Front. Neural Circuits* **7**, (2013).
7. Soetedjo, R., Kojima, Y. & Fuchs, A. Complex spike activity signals the direction and size of dysmetric saccade errors. in *Progress in Brain Research* vol. 171 153–159 (Elsevier, 2008).
8. Ohmae, S. & Medina, J. F. Climbing fibers encode a temporal-difference prediction error during cerebellar learning in mice. *Nature Neuroscience* **18**, 1798–1803 (2015).
9. Bouvier, G. *et al.* Cerebellar learning using perturbations. *eLife* **45** (2018).
10. Shankar, K. H. & Howard, M. W. A Scale-Invariant Internal Representation of Time. *Neural Computation* **24**, 134–193 (2012).
11. Chabrol, F. P., Arenz, A., Wiechert, M. T., Margrie, T. W. & DiGregorio, D. A. Synaptic diversity enables temporal coding of coincident multisensory inputs in single neurons. *Nature Neuroscience* **18**, 718–727 (2015).

REVIEWERS' COMMENTS

Reviewer #1 (Remarks to the Author):

I had only relatively straightforward expository comments on the first submission which have been thoroughly addressed in the revision.

Reviewer #2 (Remarks to the Author):

The authors have done a good job addressing the raised concerns. And I believe this paper comprises a significant contribution to our understanding of cerebellar function and timing.